# In situ light-field imaging of octopus locomotion reveals simplified control

Kakani Katija[1,2 ✉], Christine L. Huffard[1,2], Paul L. D. Roberts[1,2], Joost Daniels[1,2], Jon Erickson[1], Denis Klimov[1], Henry A. Ruhl[1] & Alana D. Sherman[1]

Animals have developed many different solutions to survive, and these abilities are inspiring technological innovations in a wide range of fields including robotics[1–3]. However, biologically inspired robots, especially those mimicking octopus locomotion[4,5], are based on limited in situ behavioural data owing to the complexity of collecting quantitative observations. Here we describe deployments of a remotely operated vehicle, equipped with a suite of imaging systems, to study the mechanics of locomotion in the octopus *Muusoctopus robustus* at the recently discovered 3,000-m deep Octopus Garden. Using a recently developed light-field imaging system called EyeRIS and an ultra-high-definition science camera, we were able to capture wide and zoomed-in views to characterize whole-animal gaits in a completely unconstrained environment across multiple individuals. Furthermore, the real-time volumetric data captured using EyeRIS yielded quantitative kinematics measurements of individual octopus arms during crawling, showing regions of high curvature and strain concentrated at distinct arm locations. Our results indicate that *M. robustus* crawling patterns showed several elements of simplified control, with implications for the design of future octopus-inspired robots. Further developments and deployments of technologies such as EyeRIS, coupled with capable robotic vehicles, will enable mining of the deep ocean for biological inspiration.

Deep ocean animals have evolved a plethora of solutions to survive in an environment starkly different from our own: one that is cold, dark, high pressure and, of course, immersed in water. These environmental conditions pose substantial challenges for observation, and while the ocean represents approximately 99% of the habitable ecosystem on the planet[6], less than about 1% of this volume has been explored at biologically relevant spatial scales. These gaps in observational capacity have ultimately resulted in our limited understanding of the ecology and behaviour of most marine animal groups. This lack of understanding is especially acute for animals in exceedingly remote locations such as the deep sea. Expanding our observational capabilities and attaining this knowledge can pay dividends in how we understand ocean system function and exploitation in a time of great change. Furthermore, knowledge about these biological systems can contribute to the exploding field of bioinspired design[1,2], in which mechanistic understanding of biological systems can be applied to engineered systems to address a variety of societal needs. Numerous studies of marine animals have led to advances in optics, propulsion, robotics, and power generation for example[7–9]. By addressing our observational challenges of ocean life using advancements in vision, perception, and robotics, previously inaccessible organismal models are poised to lead to a new era of bioinspired design.

Octopuses have inspired many breakthroughs in robotics[3]; however, their complexity poses considerable challenges to emulation. They have eight radially arranged, hyperflexible muscular hydrostat arms that afford exceptional degrees of freedom[10]. Each of these appendages is lined with up to hundreds of individually controlled suckers that can simultaneously access multiple sensory channels[11] while manipulating objects[12]. Adding further to their biomechanical complexity, octopuses can undergo rapid, drastic three-dimensional shape changes[13]. They are able to navigate obstacles and rough environmental terrain, including objects that are outside their direct field of view[14]. Their arms and suckers are coordinated by a highly distributed[15,16] but uniquely interconnected peripheral nervous system[17], with the capacity to involve oversight and learning mediated by the central nervous system[18]. Yet how their neural control orchestrates whole-body biomechanical complexity without becoming overburdened is a remarkable capability that remains to be understood.

To replicate the movements and activities of octopuses in robotic systems, researchers strive to reduce the complexity of how this system is controlled[19]. Any mechanism that simplifies the movements of an octopus should help to lessen the computational load on its nervous system and, in theory, should do the same for bioinspired robots. Aquarium and modelling studies, typically focused on a single arm (sometimes severed from the body), have found that octopuses simplify their movements using self-recognition, joint-like movements, and conserved motion primitives[19–22]. While acknowledging considerable complexity of arm ultrastructure and performance, most octopus-inspired models and robots treat the arms as simple, uniformly tapering rods or cones[4,19]. On the basis of a single study of individuals in aquaria, octopuses are

[1]Monterey Bay Aquarium Research Institute, Moss Landing, CA, USA. [2]These authors contributed equally: Kakani Katija, Christine L. Huffard, Paul L. D. Roberts, Joost Daniels. ✉e-mail: kakani@mbari.org

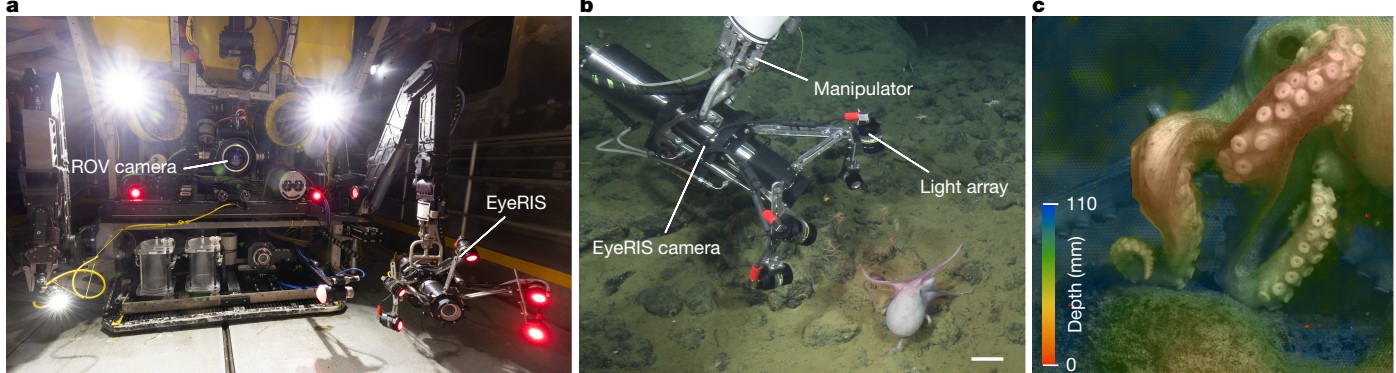

**Fig. 1 | The EyeRIS instrument as deployed at Octopus Garden. a**, The EyeRIS instrument was affixed to ROV *Doc Ricketts*. **b**, View of EyeRIS from the ROV science camera while EyeRIS was recording the locomotion of octopus O15 using a combination of red instrument lights (on an array in front of the camera housing) and white illumination from the ROV (visible in **a**). Scale bar, 10 cm. **c**, A view of a stationary octopus (O14) from the perspective of the EyeRIS instrument, showing a colourized depth map overlaid on the refocused image. The colour scale indicates the distance from the closest resolvable plane.

thought to lack a central pattern generator[22], which vertebrates and invertebrates commonly use to produce rhythmic movements, including during locomotion[23].

Biomechanics studies of whole octopuses in their natural environment remain rare and largely descriptive[14,24], and can provide important perspectives on how these animals operate their complex bodies effectively. For example, a recent paper[25] found that motion primitives, which have been considered fairly stereotyped on the basis of laboratory studies, were more flexible when used by free-living individuals accomplishing tasks that required rapid, dynamic precision. Biomechanical studies of whole-animal movements are predominantly conducted in controlled laboratory environments with a suite of sensors that include markers and imaging. Although there are practical reasons for conducting observations in the laboratory, free-living individuals must simultaneously negotiate a wide variety of conditions, including unpredictable changes in terrain, water currents, and interactions with other animals. Octopuses have considerable behavioural, neurological, and ecological complexity[26] but habituate quickly in captivity[27] where they are typically housed in aquaria sized to prevent injury[28], which often limits crawling to a few strides. The ability to conduct quantitative studies using imaging with unconstrained octopuses in uncontrolled situations is an important step towards understanding their entire movement repertoire.

Imaging has long been used to document animal diversity and density in the wild[29], and researchers are increasingly turning to imaging to study fine-scale organismal biomechanics in the field. These approaches include multi-camera measurements, in situ fluid–structure interaction measurements using dyes and specialized illumination systems, and structure-from-motion or laser scanning to generate three-dimensional reconstructions of animal morphology[30–32]. Although powerful, these approaches lack the ability to quantify time-varying three-dimensional movements, which are required to elucidate complex phenomena such as octopus locomotion. Approaches such as tomographic particle image velocimetry and holography hold promise, but have had limited application to underwater observations of marine life at the required spatiotemporal scales[33,34]. Using these approaches to study octopuses will be challenging because these animals can be notoriously difficult to find and follow in the wild, and their body postures can obstruct views of arms involved in locomotion. To construct a three-dimensional view of the movements of free-living octopuses, technologies that facilitate these observations are needed.

Here we present the design of a light-field imaging system, called EyeRIS (Remote Imaging System), which enables three-dimensional imaging and visualization in a compact payload that can be integrated onto deep-diving underwater robotic vehicles. We demonstrate the utility of this instrument by studying crawling by octopuses at the Octopus Garden on Davidson Seamount (California, USA, 3,200-m deep), where *M. robustus* can be found reliably in high densities[35] displaying a variety of behaviours (for example, brooding, crawling and swimming). Using EyeRIS, we were able to non-invasively analyse organismal features and measure the three-dimensional trajectories of distinct arms for several octopuses while they crawled over rough terrain. This approach identified a number of potential mechanisms for simplified control that can be valuable for designing octopus-inspired robotic systems: the concentration of strain and bending in conserved regions along weight-bearing arms, and consistency in gait attributes across individuals. Measurements like those enabled by imaging systems such as EyeRIS can transform our ability to understand the biomechanics and behaviour of marine life. When coupled to highly manoeuvrable robotic platforms, they can expand horizons and increase the number of available organismal models for bioinspired design research.

An expedition to Davidson Seamount, which is located 100 km west of the central California coast in the Monterey Bay National Marine Sanctuary, was conducted in August 2022 using the 4,000-m rated remotely operated vehicle (ROV) *Doc Ricketts*, which was deployed from RV *Western Flyer* (Fig. 1 and Extended Data Fig. 1). This large seamount hosts a variety of habitats, including coral reefs and sponge gardens, from its base at around 3,500 m to its crest at 1,250 m. On the southeast side of the seamount (Octopus Garden; 35.518911 °N, 122.64085 °W; approximately 3,230 m deep), one of the largest aggregations of *M. robustus* octopuses can be found, consisting primarily of brooding females (demonstrated by an inverted body posture with arms continually manipulating eggs in the clutch)[35]. In addition to brooding females, this site also hosts males and females crawling on the rocky substrate. Given the size of the aggregation, consistent presence of the animals, the slow pace of animal movement in ambient temperatures of approximately 1.6 °C, and the suite of octopus behaviours that can be observed, this site was a prime location to study octopus locomotion in the deep sea using EyeRIS. Footage from the ROV (Fig. 1a) science camera and EyeRIS (Fig. 1b) were collected independently or simultaneously, and a subset of these data with crawling octopuses was chosen for further morphometric and biomechanical analyses (individuals O1, O3, O14 and O15 in Table 1) to understand the control of weight-bearing arms during crawling.

During a single ROV dive to Octopus Garden, morphometric and biomechanical observations using either EyeRIS or the ROV science camera were conducted on 12 individuals, including both males and females (Table 1 and Supplementary Video 1). Morphometric measurements (Fig. 2) yielded eye (Fig. 2b) and head widths (Fig. 2a), verified by the quality of three-dimensional surface data (Fig. 2c), ranging from 12.3 to

## Table 1 | Analyses of individual octopuses

| ID | Eye width (mm) | Head width (mm) | Sex | Gait analysis | Kinematics analysis | Behaviour |
|---|---|---|---|---|---|---|
| O1 | NA | NA | Male | Yes[a] | No | Crawling |
| O2 | NA | NA | Male | NA | No | Swimming |
| O3 | 15.5±0.5[a] | 68.3±5.4[a] | Female | Yes[a] | No | Crawling |
| O4 | 11.7±0.5[a] | 58.7±3.0[a] | Male | NA | No | Swimming |
| O5 | 12.6±1.0[a] | NA | Female | NA | No | Swimming |
| O6 | 13.3±0.9[a] | 64.7±3.9[a] | Male | No | No | Crawling |
| O7 | 15.6±0.8[a] | 83.1±5.8[a] | Female | No | No | Crawling and swimming |
| O8 | 16.8±1.1[b] | 90.3±1.4[a] | Male | No | No | Crawling and swimming |
| O9 | 16.3±1.4[b] | 52.2±0.7[b] | Female | No | No | Crawling |
| O10 | 14.2±0.3[b] | 55.6±0.5[b] | Female | No | Yes[b] | Crawling |
| O11 | NA | NA | Female | NA | No | Brooding |
| O12 | 13.1±0.5[b] | NA | Female | No | No | Crawling |
| O13 | 15.2±0.5[b] | 53.6±4.3[b] | Male | No | No | Stationary |
| O14 | 12.3±0.7[b] | 56.6±1.3[b] | Male | Yes[a] | Yes[b] | Crawling |
| O15 | 12.3±1.2[b] | 56.9±0.8[b] | Male | Yes[a] | Yes[b] | Crawling |

Octopuses were observed at Octopus Garden (Davidson Seamount). Eye and head width measurements were determined using EyeRIS data (Fig. 2) or ROV video using laser dots with a spacing of 50 mm as reference. Values are mean ± s.d. of at least three measurements from different perspectives. The sex of individual octopuses was determined by watching multiple perspectives and identifying the easily recognizable mating arm (R3) of males. Gait analysis of the whole animal and kinematic analysis of individual were measured only in crawling animals, although a suite of was observed. NA, not available.
[a]Measurements from ROV camera footage.
[b]Measurements derived from EyeRIS data.

15.5 mm and 52.2 to 90.3 mm, respectively. The largest individuals, O7 and O8, were initially crawling and later started swimming away from the ROV and into the water column. Aside from octopuses O7 and O8, the relatively small range in sizes suggests that the octopuses at the site had similar life histories at the time of our observations. In addition to observing brooding behaviours in females, individuals of both sexes were seen crawling or swimming at the site. As with previous studies presenting ROV observations of deep-sea octopuses, the presence of bright lights is acknowledged but behaviours are considered to be part of the typical repertoire of the animals[36]. Results presented here do not include overtly evasive manoeuvres or fast escape[37].

A series of crawling arm cycles (or strides) by four *M. robustus* individuals across all arms (see Fig. 3a for arm designations) were examined in the footage from the ROV camera (individual O1: $n$ = 25 strides; individual O3: $n$ = 73 strides; individual O14: $n$ = 62 strides; individual O15: $n$ = 67 strides). These data yielded phase diagrams (Fig. 3b) indicating when the specific arm was touching the substrate (black) or not touching the substrate (white) and available EyeRIS data for quantitative kinematic analyses are shown in red boxes. Arm pair 1 in particular was used less often for support and was primarily used to investigate the upcoming substrate conditions. Arm pairs 2–4 were predominantly used for weight bearing during crawling.

Data used to generate phase diagrams were then used to compute the duration of the gait cycle and percentage of time for the crawling arm cycle, during which the arm contacted the substrate (called the duty cycle) for each left–right combination of arms and individuals (Fig. 3c,d). We found minimal left–right differences in stride characteristics for the ventral-most arms (R3, L3, R4 and L4) for most individuals, and a large variability in both gait cycle duration and duty cycle for the dorsal-most arms (R1, L1, R2 and L2). For all except one individual, the arm cycle duration was longer and the contact duty cycle smaller for dorsal arms than for ventral arms.

Because multiple camera angles provided good visibility of the left and right sides of octopus O15, the sequence of its gait could be examined as it crawled forwards over rough terrain for nearly 12 min (Fig. 3b,e). Arms on both sides of the body were used for support in 73% of video frames analysed, and an average of three arms (3 ± 1) contacted the substrate at any given moment. Although the number of arms used for support during crawling was broadly consistent, this number cycled between 2 and 6 arms during periods of 2–3 min (Fig. 3e). The order of arms used in crawling (Fig. 3f) shows a higher occurrence of cyclical contact and weight bearing from ventral to dorsal arms, with little to no occurrence of arms 1 after a consistent arm pair.

To gain insights into the fine-scale movement of octopus arms during crawling, detailed kinematics were measured directly from EyeRIS (Fig. 4 and Supplementary Video 2). We tracked points (red dots; Fig. 4a, top) on arms L2 and L3 for individual O15 as the animal straightened and stretched the proximal section of its arms before release from the substrate (at $t$ = 0 s). After release from the substrate, L3 contracted the proximal section of its arm as it cycled under the body of the animal for the next arm cycle. The location of arm bending occurred around the point of contact with the substrate, and seemed to be conserved throughout the crawling cycle. The tracked points and corresponding

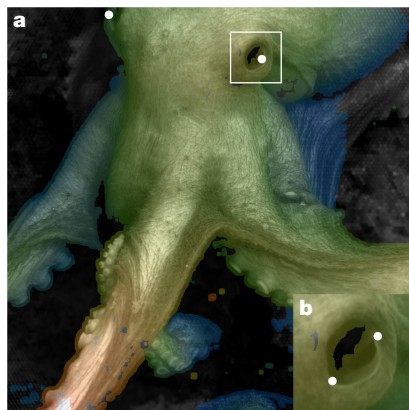
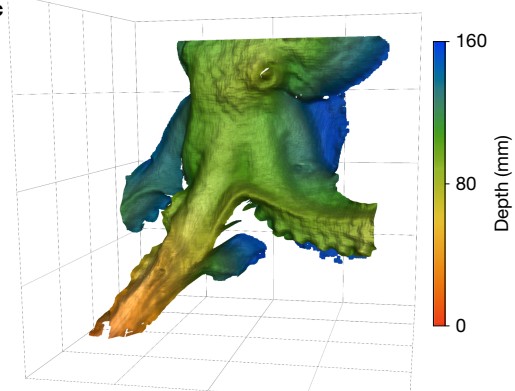

**Fig. 2 | Three-dimensional size measurement capabilities of EyeRIS.**
**a**, The head width of an octopus (O15) was measured as the distance between the eyes at its widest point; the white dots indicate the points of measurement. **b**, Eye size was measured as the largest visible eye diameter. **c**, A real-time three-dimensional model can be interrogated to verify the quality of the surface data. Colour scale indicates the distance from the closest resolvable plane. Grid spacing is 50 mm.

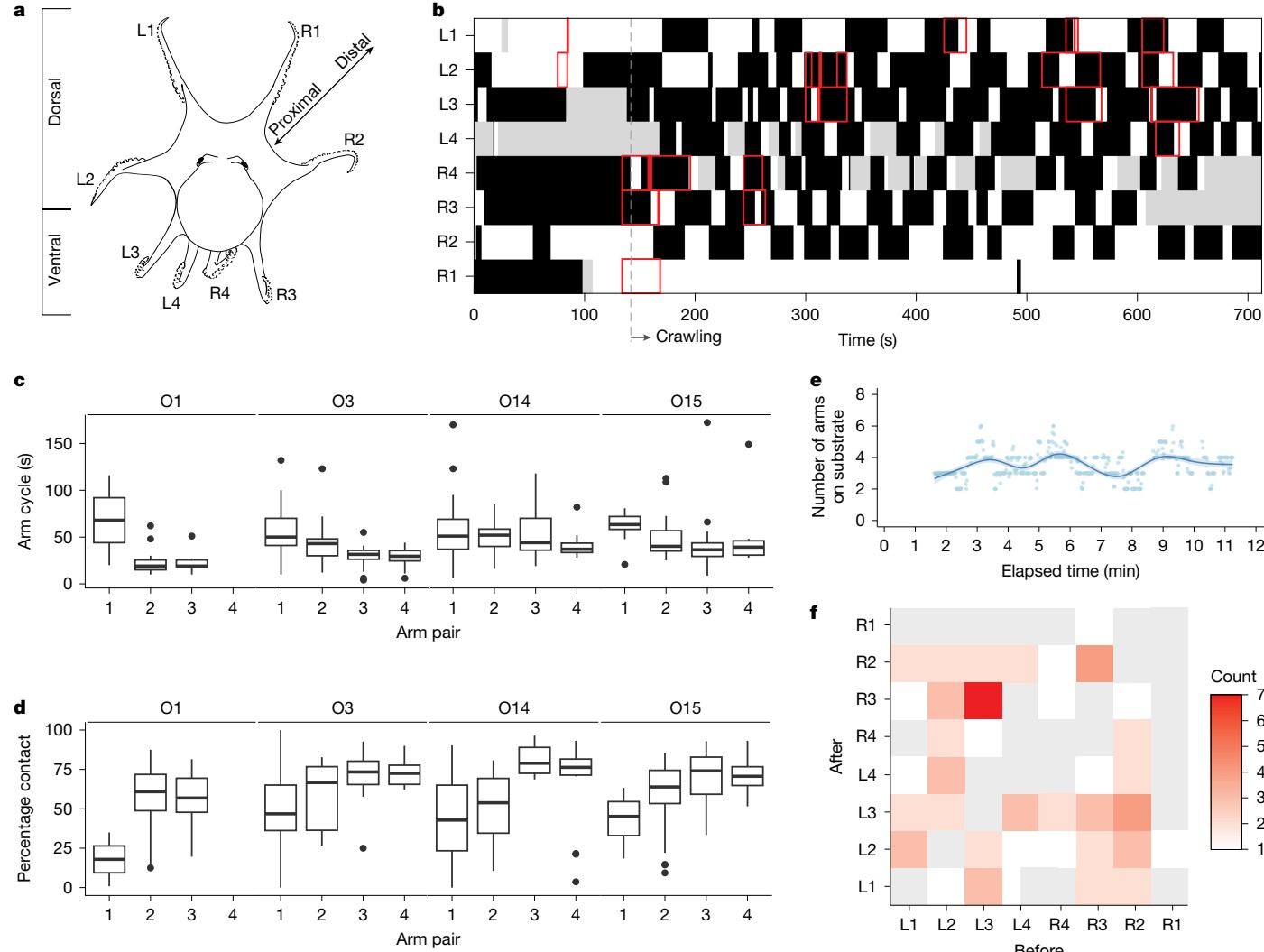

**Fig. 3 | Gait characteristics of crawling *M. robustus*. a**, Definition of arm designations. L, left; R, right. **b**, Phase diagram of crawling by individual O15 for a single 12-min observation, indicating when the arm was touching the substrate (black) or not touching the substrate (white). Grey areas indicate that the state of a given arm could not be ascertained from the various video angles. Red markers indicate when kinematic analyses were conducted using EyeRIS data. **c**,**d**, Arm cycle durations (**c**) and contact duty cycles (**d**) for each arm pair of four individuals (median values indicated in boxes; box denotes the interquartile range (IQR); whiskers denote Q1 − 1.5 × IQR and Q3 + 1.5 × IQR; dots denote data points beyond the whisker range). **e**, Number of arms on the substrate for periods when arms on both sides of the body were used for support (O15 only). Solid line and filled region show the 'gam' smoothing line and 95% confidence interval, respectively. **f**, Order of arms used by O15 during crawling (during the sequence in **b**). *y* axis shows the arms that touched the substrate immediately after the arms on the *x* axis were used (grey, no occurrence).

depth maps (Fig. 4a, bottom) provided sufficient information to plot the positions of the tracked points and interpolated splines describing three-dimensional arm movements (Fig. 4b); line colours indicate the time. Furthermore, we identified regions of high curvature (or bend radius; Fig. 4c) and high strain (or stretch; Fig. 4d). Fine-scale kinematics analyses on other individuals and arms can be found in Extended Data Figs. 2–4. Aggregate results for curvature and strain for individuals O14 and O15 and dorsal arms L1, L2 and R1 over multiple arm crawling sequences (Extended Data Fig. 5) are shown in Fig. 5a,b, respectively; results showing curvature and strain along ventral arms L3, L4 and R3 are shown in Fig. 5c,d, respectively.

## Discussion

This work presents three-dimensional measurements of octopus bio-mechanics of free-living individuals in the wild, investigates octopus moving over rough terrain, and provides some of the longest quantitative observations of unconstrained octopus locomotion (more

than five crawling strides). Our results reveal that *M. robustus* initiated each forward-crawling cycle by placing a medial portion of an arm on the substrate; for simplicity, we call this portion of the arm the foot. The octopuses maintained a region of high curvature (or bend) proximal to the foot while the body moved forward, and the proximal, weight-bearing portion of this arm changed length (Fig. 4a,b). Typically, the foot maintained its position on the substrate and was not released until the proximal portion of the arm reached maximal strain. After this point, the suckers lost purchase, and the foot was pulled off the substrate as the body continued moving forwards. While the arm was in contact with the substrate, regions of high curvature and strain were not evenly distributed along its length or even along the weight-bearing portion (Fig. 5a,b). Bend was consistently concentrated at the foot (Fig. 5a), and changes in strain were concentrated along the third of the arm immediately proximal to the foot (Fig. 5b). By contrast, the arm portion distal to the foot typically showed little change throughout the arm cycle. These conserved regions for high strain and curvature along octopus arms during crawling could yield

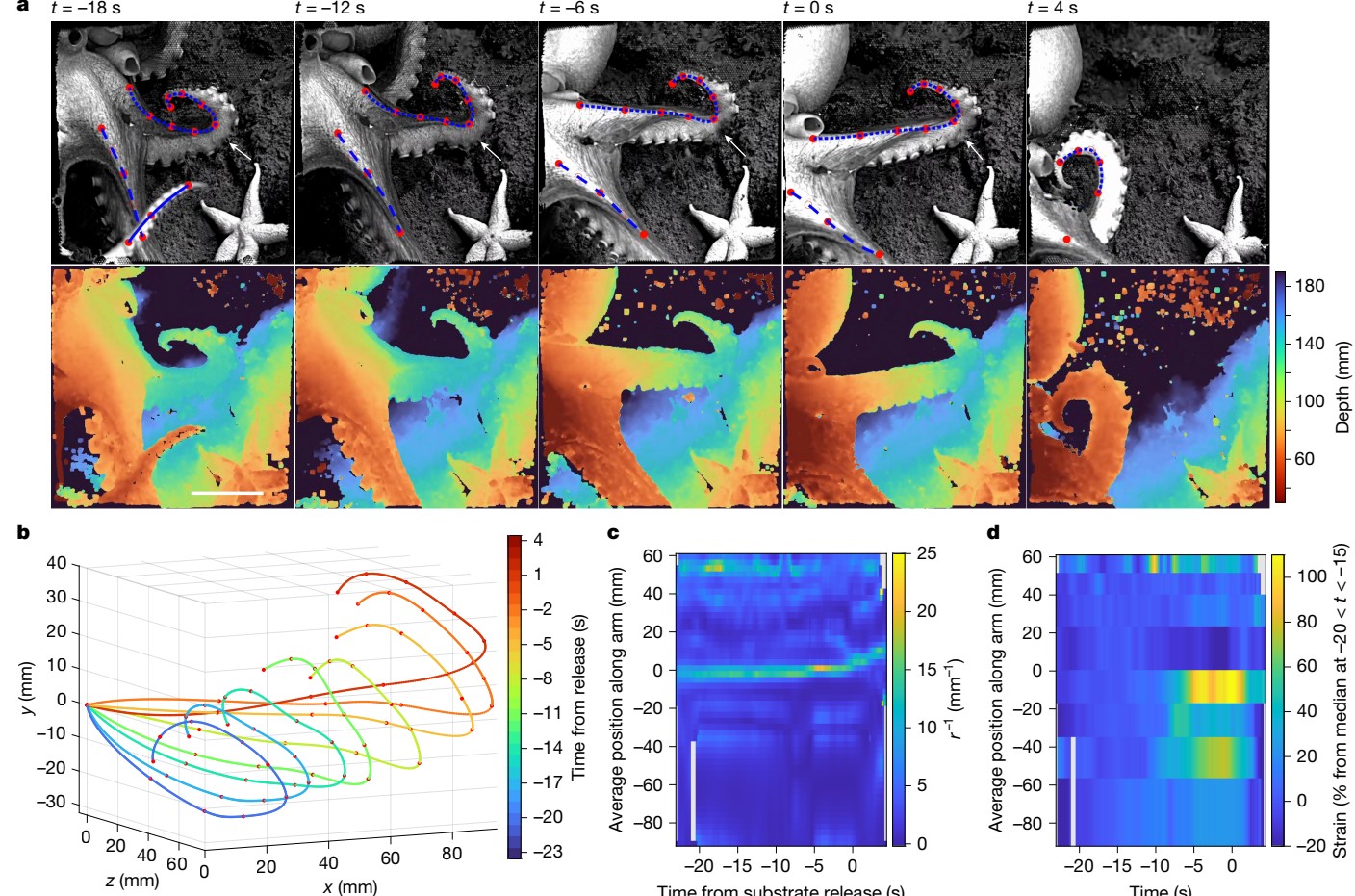

**Fig. 4 | Detailed kinematics of arm L3 of octopus O15 during crawling, where $t = 0$ s corresponds to when the arm is released from the substrate.** **a**, A crawling sequence captured using EyeRIS, showing the orthographic total focus (top) and orthographic depth map (bottom) views. Warmer colours indicate objects closer to the image sensor. Red dots indicate points tracked over time; blue dashed lines indicate the interpolated fit for each arm; white arrow indicates where the arm attaches to the substrate. Scale bar, 50 mm. **b**, The three-dimensional position of arm L3 during the crawling cycle relative to the most proximal tracked point. The position of the arm is drawn at 3-s intervals (from bottom left to top right); colouration of each segment indicates time from substrate release of arm. **c**,**d**, Time-varying curvature ($r^{-1}$) (**c**) and time-varying strain (%) (**d**) along arm L3 during the same crawling sequence. The reference position corresponds to where the arm attaches to the substrate. Warmer colours indicate regions of higher curvature and strain in **c** and **d**, respectively.

simplified implementations for octopus-inspired robots with similar performance characteristics.

The crawling patterns of *M. robustus* showed several elements of simplified, but flexible, control. Control is simplified by using joint-like motion primitives that concentrate strain in specific arm regions. Flexibility is offered by the use of feet that maintain stability as the individual traverses unpredictably rough terrain, even though the octopus is unable to see and precisely aim foot placement. Motion primitives involved in forward crawling seemed to differ from those used during backward crawling and walking by other octopuses; rearward locomotion described thus far uses a propagating bend motion primitive to produce a rolling gait[14,24,38]. Instead, forward crawling strides by *M. robustus* involved a deforming, weight-bearing portion of the arm that meets the sea floor at a functional, jointed foot that generally stayed stationary until lifted off bottom. We observed shortening, and then elongation, of this proximal portion of the arm, as was observed for *Octopus vulgaris* crawling on the side of an aquarium[22]. However, this differential shape change was not distributed evenly along the arms of *M. robustus*; some portions of the proximal arm stretched more than others. Bends were also highly concentrated in distinct regions of the arm, forming joint-like elements that may have reduced the degrees of freedom of the movement, as occurs during fetching by *O. vulgaris*[21]. Crucially, this shows that

weight-bearing and non-weight-bearing portions of the same arm do not concurrently experience equal strain during crawling. This specialization of arm portions may reflect differences in arm muscle anatomy and performance along the length, as found in *O. vulgaris*[39]. This movement may involve an underwater spring-loaded inverted pendulum, as modelled for an underwater soft robot inspired by bipedal locomotion in the coconut octopus (*Amphioctopus marginatus*)[5]. As *M. robustus* crawled over the rocky terrain, they typically used three arms at a time, suggesting that they used a centre of mass projection in a polygon of support in conjunction with, not instead of, a possible spring-loaded inverted pendulum. Like *Octopus sinensis*[40], *M. robustus* in the wild tended to use arm pairs 2–4 for locomotion and arm pair 1 less often.

Three-dimensional and underwater two-dimensional imaging methods have formed the foundation of studies identifying motion primitives in octopuses. Previous studies have used stereo cameras in the laboratory to reconstruct three-dimensional spline-smoothed trajectories of points along a single arm and model normalized velocity profiles[41]. Such approaches are especially useful for identifying conserved patterns in highly repeatable, stereotyped movements, but calculates average strain along the entire arm length. In situ two-dimensional camera work has been used to study rotation and translation of single arms as octopuses dynamically aim a motion

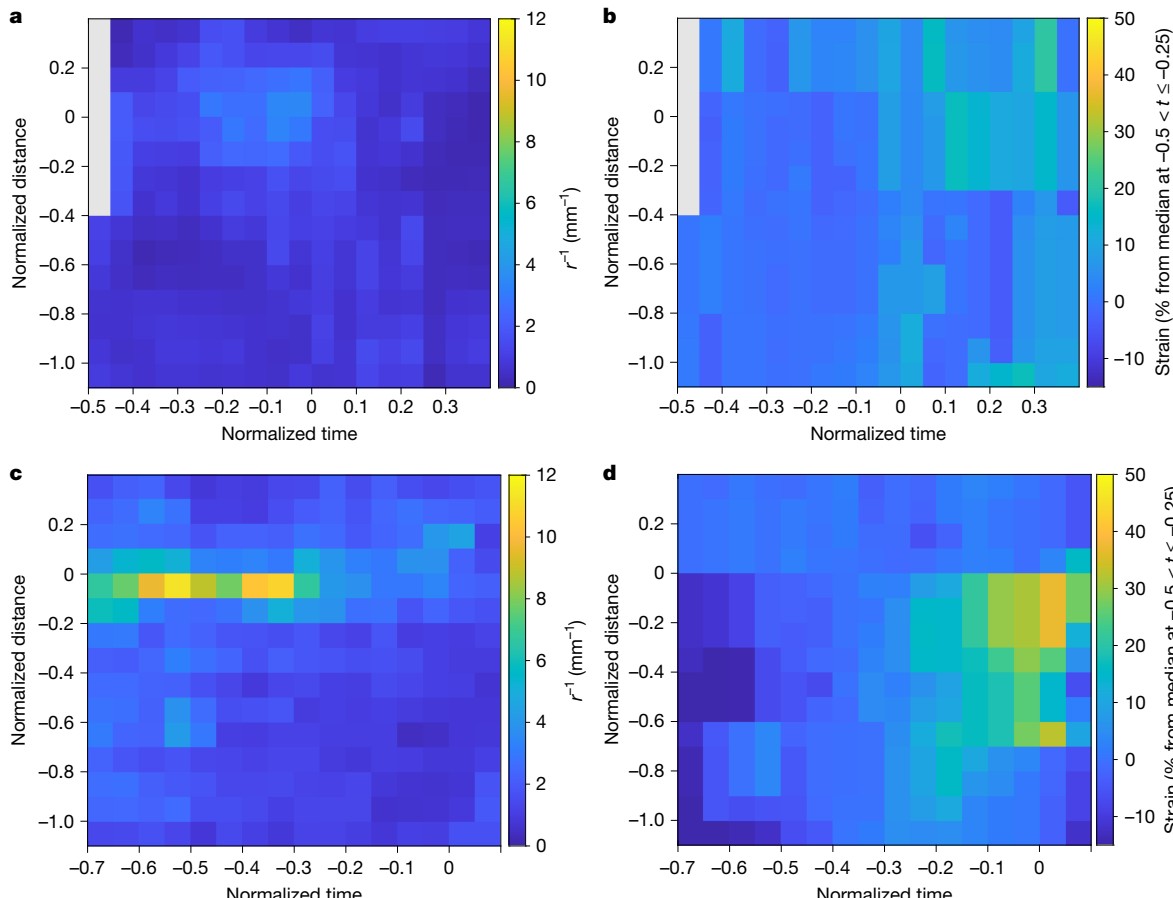

**Fig. 5 | Aggregate kinematics of arm movement over multiple crawling cycles for octopuses O14 and O15. a,b**, Time-varying curvature ($r^{-1}$) (**a**) and time-varying strain (%) (**b**) along dorsal arms L1, L2 and R1. **c,d**, Curvature (**c**) and strain (**d**) along ventral arms L3, L4 and R3. The distance along the arm is normalized to the head width of each individual; the time during the arm cycle is relative to release from the substrate and is normalized using the median arm cycle (Fig. 3c). The number of sequences used in each data bin is shown in Extended Data Fig. 5.

primitive (slapping motion) towards a moving target (a fish)[25]. Here we synthesize results from three-dimensional EyeRIS measurements and an ultra-high-definition camera system to understand volumetric changes along the arm length over time in whole animals navigating a complex environment. The approach used here illustrates how a joint-like motion primitive develops and evolves in a hydrostatic skeleton during weight-bearing locomotion. The ROV videos provided an additional perspective by revealing flexibility in gait characteristics over multiple strides (Fig. 3c,d) as the animals navigated rough terrain.

Despite moving over rocks of irregular sizes, *M. robustus* strides showed consistency, even across individuals. Ventral arms showed similarly shorter arm cycles and larger contact duty cycles than did dorsal arms (Fig. 3c,d). This consistency in stride timing may control a cyclical pattern of bilateral arm use (Fig. 3e). The recently identified sinusoidal locomotion in *Octopus bimaculoides* also shows a cyclical pattern. In that case, sinusoidal directionality during prey approach is thought to allow octopuses to achieve lateral motion parallax during visual attack[42]. The similarities in these patterns are intriguing, and future work is required to assess whether cyclical arm use is a biomechanical mechanism that controls sinusoidal locomotion.

There is an extreme paucity of biomechanics observations of live octopuses, especially in situ, compared with the large number of published studies on octopus-inspired robotics. A conservatively restrictive Google Scholar search of 'octopus inspired robot' yields 2,160 results. By contrast, searching for 'octopus locomotion' yields 27 results, of which only 4 refer to kinematic analyses of whole octopuses. We are aware of only five previous reports of octopus crawling or walking

kinematics[14,22,37,40,42] that we could compare our results to. Importantly, previous work[22,40,42] observed animals crawling against a flat, smooth-glass aquarium, although in two of the studies[40,42], the aquaria were only slightly larger than the arm span of the octopuses. Such an experimental set-up limited how many strides the animal could take before a change in direction was required. Observations presented here, by contrast, were conducted in the wild, where the unconstrained scale and variability of conditions allowed free-living octopuses to express a broader range of simultaneous control solutions. For example, the cyclical pattern of bilateral arm use to navigate uneven terrain might hint at a previously undescribed central pattern generator in octopuses. Addressing these knowledge gaps will be critical to the development of robots capable of performing actions successfully in real-world situations, using a diversity of octopus gaits[43].

In addition to the scientific contributions of whole-octopus locomotion, here we describe the successful design, build, and deployment of EyeRIS (see Extended Data Fig. 7 and Extended Data Table 1 for details on calibration and error). Our effort demonstrates that light-field imaging can be used for biomechanics studies of underwater animals in the wild. Although body sizes of deep-sea octopuses have been estimated from video using sizing lasers[44], detailed morphometric measurements have typically required examination of dead specimens by hand[45] using calipers[46]. Being able to use quantitative imaging techniques such as EyeRIS can enable non-invasive measurements that inform species descriptions of rare and/or less-understood soft-bodied animals[47]. The potential applications for light-field imaging are immense: beyond morphometric and kinematics studies on other benthic and

midwater animals (Extended Data Fig. 6), the technique can be used to generate time-varying surface reconstructions[48,49] and/or simultaneous three-dimensional flow field data (by measuring the time-varying displacement of suspended particles) to inform a plethora of animal–fluid and fluid–structure interaction problems in situ. Although this imaging approach would be challenged considerably by highly turbid environments, with appropriate illumination and housing hardware, it could be used to analyse a number of areas of inquiry ranging from near-surface to deep-sea environments.

Finally, the importance of in situ biological observations and their quantitative study is not only useful for the improvement of our understanding of these systems as a whole, but informs many downstream applications as evidenced by the field of bioinspired design. Despite the popularity of the field[1,2], there is a disappointing lack of successful bioinspired technologies used at industrial scales[50]. This could be due to many reasons, other than the lack of technological approaches that enable quantitative observations in situ, including: (1) poor interpretation of published biological literature; (2) poorly defined and subsequently limited use of robust and indicative quantitative metrics that translate well to engineering design; and (3) distant (or absent) collaborations with biological subject-matter experts who are intimately familiar with organismal function, ecology, and other areas[51]. Whatever the reason, there remain a number of areas for improvement in quantitative observations of biomechanics in the wild that will continue to challenge the broader community.

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

## Methods

The imaging data presented in this paper were collected using an underwater light-field imaging system called EyeRIS and an ultra-high-definition 4k ROV science camera. Both imaging systems were integrated on ROV *Doc Ricketts*, and all systems are rated to operate at depths of up to 4,000 m (Fig. 1). The camera is mounted on a pan–tilt mechanism that is affixed to the core vehicle. EyeRIS is mounted on the manipulator arm of the ROV, which enables fine control and positioning of the instrument depending on the scientific question we focused on.

### EyeRIS

EyeRIS uses light-field (or plenoptic) imaging technology. At a fundamental level, light-field imaging requires taking images of a scene from several viewing angles[52,53]. It is generally grouped into two categories: single-camera light field and multi-camera light field[54]. Here we focus on the former owing to size and resource constraints for deployment in deep underwater environments. Several viewing angles are obtained by placing a microlens array in front of a regular two-dimensional image sensor. If the microlens is placed one focal length from the front of the imager, then each microlens images the incoming ray angle as pixels[55]. By contrast, if the microlens array is placed at a distance larger than one focal length from the microlens array, the microlenses map the virtual three-dimensional scene onto the sensor with groups of microlenses providing a collection of different perspectives of the same patch in the scene[56,57]. The second approach is what we use here. In either case, software is used to transform the raw light-field data into depth maps and focused two-dimensional images. The depth map is then used to produce an orthographic total focus image for which the spatial scale is independent of the depth in the scene. From these data, three-dimensional surface reconstructions and point measurements can be derived from a single-camera perspective.

### System overview

The EyeRIS light-field imaging system (Extended Data Fig. 1) was built around a Raytrix R26 camera (Raytrix). This camera has a 25-megapixel resolution, and records up to 80 frames per s with a 4.5-μm pixel resolution. The main camera lens is a Canon 70–200-mm zoom lens with f/2.8 aperture. Most of the system is housed in a single pressure case that is machined from a solid grade-5 titanium bar and rated for 4,000 m depths. A flat BK7 view port with an anti-reflection coating on the interior of the port provides the optical interface between the main camera lens and seawater.

A CoaXPress to fibre extender PHT4 (Phrontier Technologies) was used to convert the four-channel CoaXPress output from the R26 to four wavelengths carrying the camera image signals and one wavelength carrying the camera-control signal. These five wavelengths were multiplexed together using a custom-built coarse wavelength division multiplexing system provided by the manufacturer of the PHT4 module. This resulted in one, single-mode fibre output from the instrument for camera data and control that could be transmitted through the ROV umbilical cable. The single-mode output was connected to the ROV using a Greene Tweed ST-DRY bulkhead rated for 10,000 PSI and a Linden Photonics subsea patch cord. Using the fibre for camera data and control, data rates of 12 Gb s$^{-1}$ enabled data viewing at 60 frames per s with minimal lag.

Power and 100 Mb ethernet were provided for instrument control from the ROV using a Seacon MINK10/L connector and cable. The system used two Bel Fuse power supplies with universal a.c. and high-voltage d.c. input so that the power inputs were as flexible as possible. This enabled operation at typical US and European a.c. voltages and at up to 390 V d.c. One power supply (MBC275) provided up to 275 W for external lights; the other (MBC41) supplied power to the rest of the EyeRIS components. A Raspberry Pi 3 (Raspberry Pi Foundation) single-board computer was used to control the power relays that switch the camera and light power paths, the stepper motor that controlled the lens zoom setting and the interface with environmental sensors in the pressure housing that monitor temperature, pressure and humidity[58]. A Birger lens controller (Birger Engineering) was used to monitor the zoom settings on the lens. A small Python library running on the Raspberry Pi abstracted the power and lens control, sensor monitoring and system status into a set of lightweight communications and marshalling channels that could be subscribed to by any device on the network. A companion Python graphical user interface running on the light-field processing computer subscribed to these channels and visualized the sensor data in real time and provided buttons to control the state of the power relays, light levels and zoom settings.

Although the zoom lens enables a continuous range of focal lengths to be selected between 70 mm and 200 mm, light-field calibration must be performed for each individual focal length. As a compromise, a discrete set of focal lengths (70, 100, 135 and 200 mm) were selected and calibrated, and the EyeRIS camera-control software[58] was set to drive the lens only to these focal lengths. The zoom drive mechanism was built using an off-the-shelf stepper motor (Faulhaber DM1220 256:1 gear ratio) and an off-the-shelf motor controller (Pololu Tic t825)[59]. The most commonly used zoom setting during our field deployments was the widest at 70 mm.

Illumination from EyeRIS was provided by an array of five SeaLite light emitting diode lights (Deep Sea Power and Light). The light power was controlled using RS-485 communications to the Raspberry Pi. The lights were mounted on a hydraulically actuated array that uses a two-bar linkage to expand and contract the array with two hydraulic cylinders. Hydraulic power and control for the illumination array was provided by the ROV.

Raytrix provides an extensive software package named RxLive for processing and visualizing light-field data from their cameras in real time. We used this software as the primary interface to the camera for data visualization, in situ three-dimensional measurements, recording and the first step in post hoc processing. Data outputs include 'total focus' images, which resemble the image that a monochrome-perspective camera would obtain but with a large depth of field spanning the entire image volume of the light-field camera. Another data output—the depth maps—provides the depth (or range) of each pixel in the image. These are typically visualized using colour maps and can be overlaid on the total focus image to enable three-dimensional perception on regular computer screens (Fig. 1c). The four-channel images containing calibrated three-dimensional coordinates and total focus luminosity were used to derive the final data product: orthogonal projections of the square field of view for each frame. This final data product was used during quantitative analysis. In addition to the above, RxLive was used extensively for in situ control and decision-making, adjustments to the camera placement and settings and to select data for analysis.

### EyeRIS calibration and validation

Calibration of EyeRIS was conducted using RxLive with the aid of a calibration target. This typically involves a two-step process in which the microlens array is first calibrated (for example, the position of the image of each lens is identified on the sensor). Next, a metric calibration is used to estimate the camera intrinsics[57]. Both the microlens array calibration and the metric calibration are dependent on the focal length and aperture of the main lens and, to a lesser extent, the refractive index of the medium. Therefore, we performed calibrations at a discrete set of focal lengths of the zoom lens—at 70, 100, 135 and 200 mm—with the housed camera and calibration target submerged in seawater. It should be noted that typical variations in the refractive index of seawater due to pressure, temperature and salinity do not significantly change the calibration.

Once the camera was calibrated, we performed validation measurements to confirm that the calibration was successful and that

measurements made in situ would have sufficient accuracy and precision for the scientific application (Extended Data Fig. 7). The validation consisted of imaging the calibration target at various locations in the field of view of the camera, making measurements between points on the target using RxLive and comparing these measured distances to known distances on the target. These measurements were repeated over different distances and locations in the imaged volume and statistical analysis of these measurements showed that the average error was less than 2% of the distance measured (Extended Data Table 1). Typical depth maps using EyeRIS in this study have an uncertainty of less than 1 mm (Extended Data Fig. 7d).

### EyeRIS operations

Animals of interest were selected according to their behaviour and environment using the main ROV science camera. For the purposes of this study, we focused on animals that were actively crawling, in an area that enabled vehicle placement without disturbing any octopuses or causing considerable substrate damage, as well as with enough space free of obstacles to precisely manoeuvre the instrument for the best view. Once selected, the vehicle was placed on the ocean bottom, and the manipulator arm adjusted to centre one or more arms in the imaged volume of EyeRIS (Supplementary Video 1). The instrument was typically placed perpendicular to the vehicle heading, enabling a clear view of the instrument and the octopus using the ROV cameras. Once lined up, lighting was adjusted as needed, balancing white ROV lights and the red light of the EyeRIS light emitting diode array, with the latter being used mostly to fill in any shadows cast by the octopus or instrumentation. Camera shutter speed was then used to fine-tune the exposure. Owing to the high data rate of the instrument (1.5 Gb s$^{-1}$), recordings were 30 to 60 s in duration, at a frame rate of 60 frames per s. The live depth map feed was used to manually adjust the positioning of the vehicle and manipulator during recording as needed.

### Whole-octopus gait analysis

To analyse the gait at the whole-animal level, each time an arm touched the substrate was recorded. This was done manually, using multiple camera views where possible (from the ROV cameras and the EyeRIS camera). We also noted when the state of an arm could not confidently be determined from the video. From these data, the number of arms on the substrate at any given time was determined, as well as duty cycle statistics and the percentage of time each arm spent in contact with the substrate.

Since the early days of cephalopod biology, researchers have recognized the difficulty of assigning anatomical directionality to octopuses, which have eight radially arranged, but bilaterally symmetrical arms[60], and two eyes but functionally monocular vision[61]. We follow the directionality convention used by taxonomists[60] because its consistent reference to underlying morphological features is independent of the direction of movement, which is rarely in line with the longitudinal axis of the body[22] and often at odds with the direction of vision[61]. In this convention, arm pairs are identified according to their left–right symmetry, with dorsal arms as arm pair 1 and ventral arms as arm pair 4.

### Markerless tracking of octopus arms

To capture quantitative kinematics details of individual arms, processed EyeRIS data were used. Arm movement was digitized by first generating orthogonal total focus projections and depth maps from the EyeRIS data using RxLive and performing markerless point tracking on features visible in the total focus images. Tracking was performed through a combination of automated cross-correlation and manual point selection using DLTdv8[62]. Skin patterns and morphological features (for example, sucker location) were used as a reference, but adjustments were made to enable continuous tracking when arm torsion moved features out of view and to more accurately estimate overall arm attitude using a linear sequence of points. Here we distinguish

between three different types of point: key points track the longitudinal position along the arm to the extent possible; guide points were added later to improve the interpolation of the arm location between key points; and reference points track background and substrate features that were later used to correct for vehicle and instrument movement during measurements.

Using the two-dimensional point locations ($x$ and $y$ coordinates, associated depth or $z$ coordinate), data were retrieved from depth maps after smoothing using an $11 \times 11$ pixel median filter. This process yielded three-dimensional position data in real-world coordinates over time for all tracked points for which valid depth information was available in the EyeRIS light-fields images. Several steps were then taken to correct the data and improve consecutive arm position analysis: (1) a five-frame median filter was applied to the $z$ coordinate to reduce measurement noise; (2) three-axis background motion was removed using the tracked reference points; (3) outliers were eliminated and a 20-frame moving-average filter was applied to smooth the point data; and (4) small gaps (<100 frames) were filled using piecewise cubic interpolation. It is worth noting that for outlier correction, a baseline was generated by applying a moving median width of 50 time steps (0.83 s) to the $z$ coordinates. Any point more than 5 mm or one s.d. from the reference point (whichever was greater) was removed. Arm strain and curvature analysis can be sensitive to the interpolation method chosen: we opted to use a simple spline to preserve smooth bends and iteratively added tracking points as needed to constrain the fit. Furthermore, a seven-frame moving-average filter was applied to the interpolation to improve stability. The distance between tracking points was then measured along the fit for each recorded frame to quantify the strain (or stretching) of each arm section over time. Similarly, the curvature (or bend radius) was determined for regularly spaced points along the fit at each frame.

### Reporting summary

Further information on research design is available in the Nature Portfolio Reporting Summary linked to this article.

## Data availability

All data used in this work can be accessed at Zenodo (https://doi.org/10.5281/zenodo.10795493)[63]; EyeRIS design and control repositories can be found at GitHub (https://github.com/bioinspirlab/eyeris-camera-control and https://github.com/bioinspirlab/eyeris-zoom-drive)[58,59]. More details on the EyeRIS instrument can be found at https://www.mbari.org/technology/eyeris/.

## Code availability

All code used in this work can be accessed at GitHub (https://github.com/bioinspirlab/octopus-3d-tracking)[64].

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

**Acknowledgements** We benefited from conversations with J. Barry and S. Litvin about octopus behaviour at the Octopus Garden dive site, and we thank numerous technical staff members at Monterey Bay Aquarium Research Institute (including the crew members and pilots of RV *Western Flyer* and ROV *Doc Ricketts*, respectively) who provided support for EyeRIS instrument development, integration, and deployment. This work is a contribution of the Bioinspiration Lab. This work was supported by the David and Lucile Packard Foundation and the Gordon and Betty Moore Foundation (no. 7583 to K.K., H.A.R. and A.D.S.).

**Author contributions** Conceptualization, K.K., C.L.H., P.L.D.R. and J.D.; data curation, P.L.D.R. and J.D.; formal analysis, C.L.H. and J.D.; investigation, C.L.H. and J.D.; methodology, K.K., C.L.H., P.L.D.R., J.D., D.K., J.E. and A.D.S.; project administration, K.K.; resources, K.K., A.D.S. and H.A.R.; software, P.L.D.R. and J.D.; supervision, K.K. and A.D.S.; validation, K.K., C.L.H., J.D. and P.L.D.R.; visualization, K.K., C.L.H., J.D. and P.L.D.R.; writing—original draft, K.K., C.L.H., J.D. and P.L.D.R.; and writing—review and editing, all authors.

**Competing interests** The authors declare no competing interests.

**Additional information**
**Correspondence and requests for materials** should be addressed to Kakani Katija.

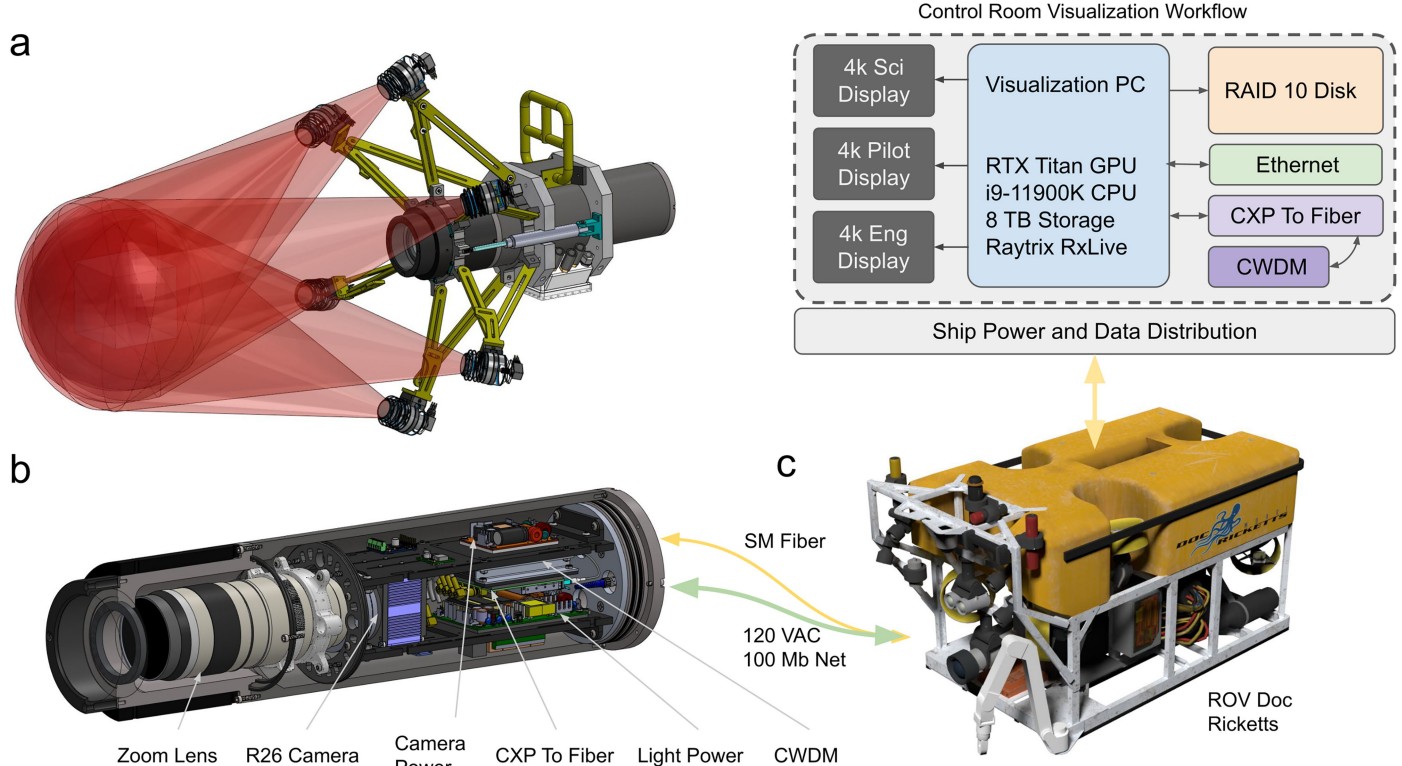

**a**

Control Room Visualization Workflow

| 4k Sci Display | Visualization PC | RAID 10 Disk |
| 4k Pilot Display | RTX Titan GPU i9-11900K CPU 8 TB Storage Raytrix RxLive | Ethernet |
| 4k Eng Display | | CXP To Fiber |
| | | CWDM |

Ship Power and Data Distribution

**b**

Zoom Lens    R26 Camera    Camera Power    CXP To Fiber    Light Power    CWDM

**c**

SM Fiber

120 VAC
100 Mb Net

ROV Doc Ricketts

**Extended Data Fig. 1 | Overview of the *EyeRIS* instrument. (a)** A computer-aided design model of the system showing the light array expanded and lighting geometry indicated by the red region. **(b)** Cutaway view of the internal components of the system that includes a zoom lens, camera, and hardware to power, control, and transmit imaging data. **(c)** System diagram of the control room user interface interface from the ship to ROV *Doc Ricketts* and *EyeRIS*.

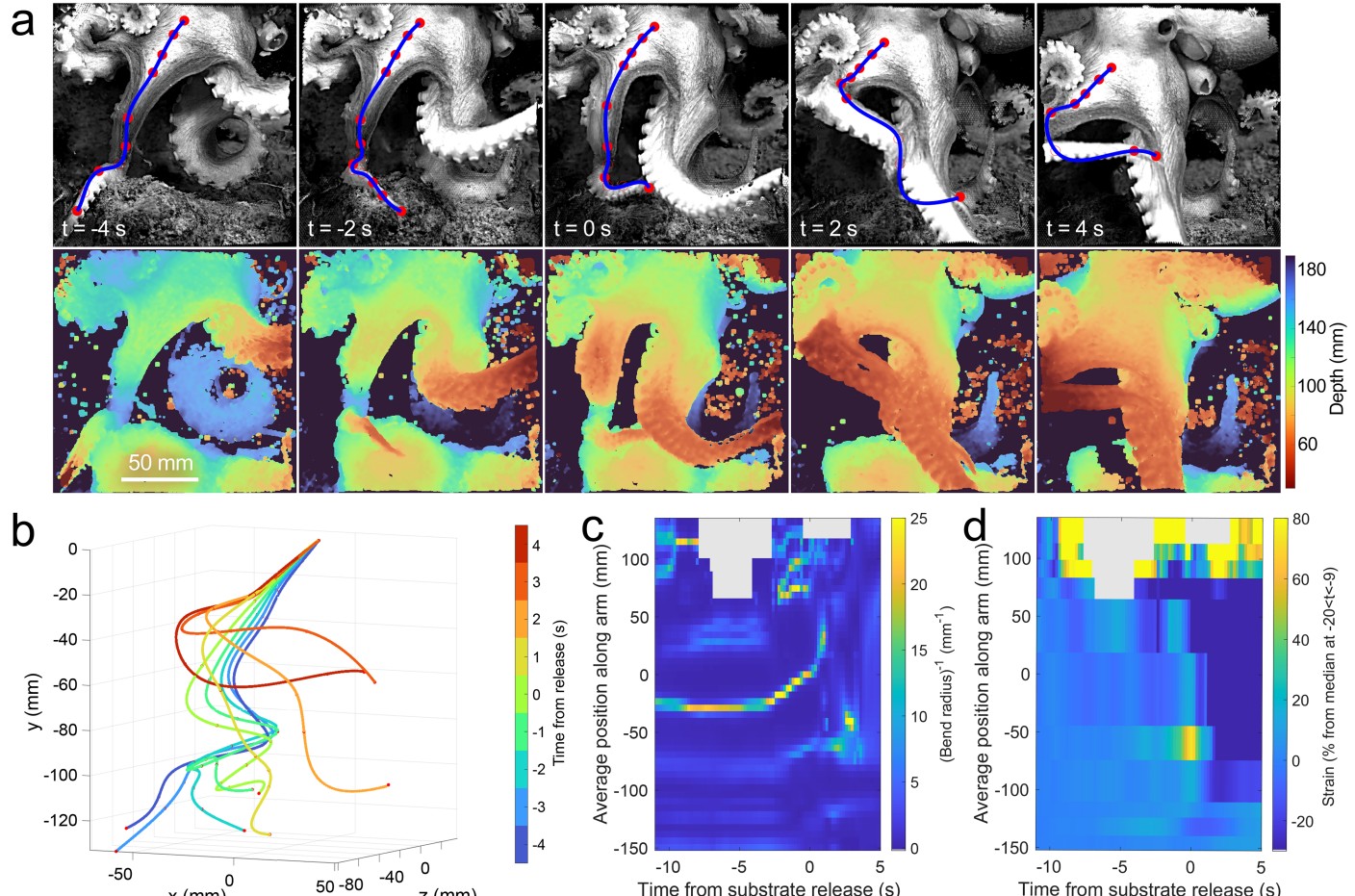

**Extended Data Fig. 2 | Representative data of the detailed kinematics of arm L1 of individual O15.** (**a**) Total focus images at various times relative to release of the contact point, with tracked points overlaid (top row) and depth maps (bottom row). (**b**) 3D plot of arm position relative to the most proximal tracked point colored by time from substrate release of the arm. (**c**) Curvature and (**d**) strain of arm L1 over time.

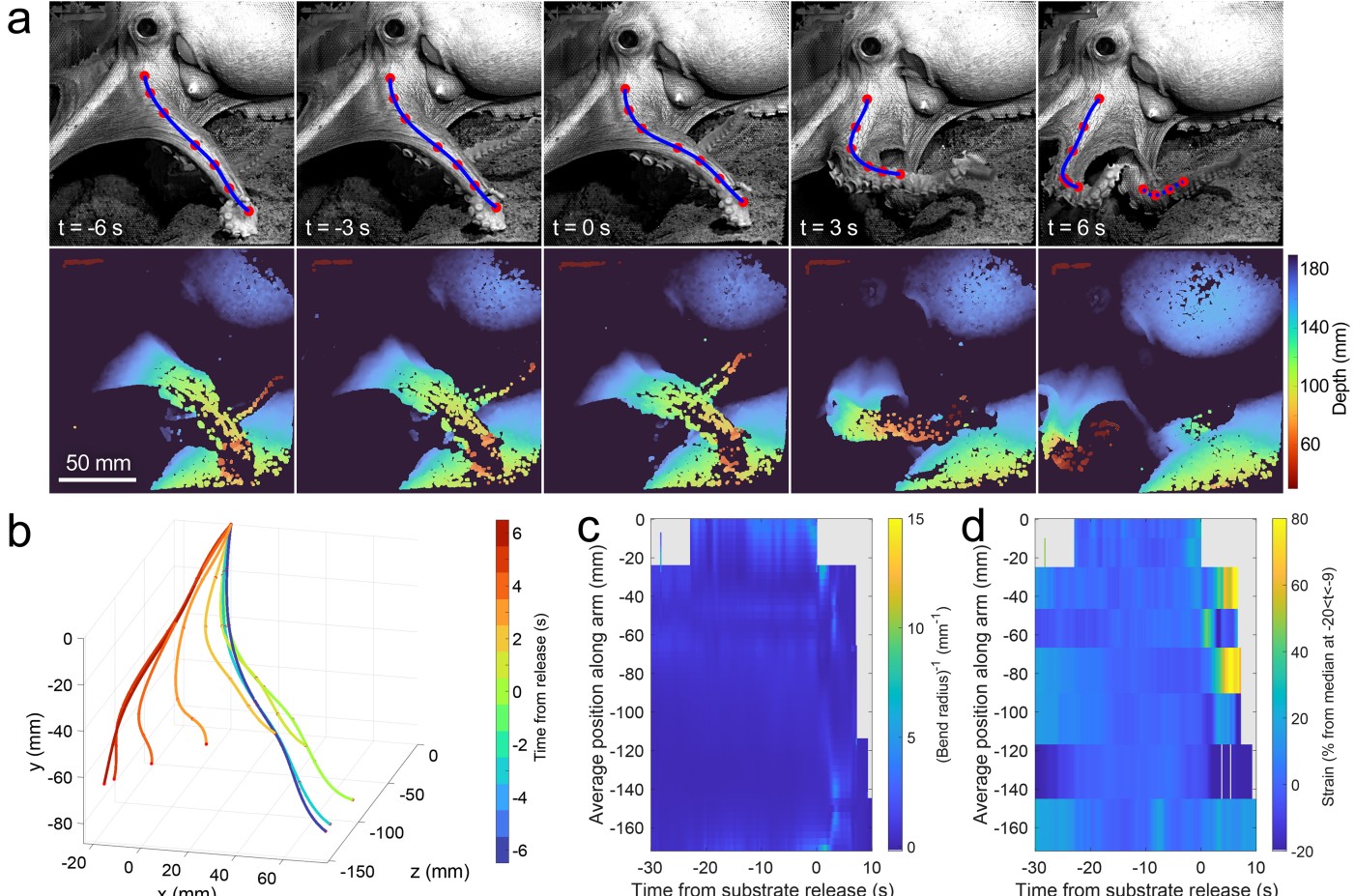

**Extended Data Fig. 3 | Representative data of the detailed kinematics of arm L2 of individual O14.** (**a**) Total focus images at various times relative to release of the contact point, with tracked points overlaid (top row) and depth maps (bottom row). (**b**) 3D plot of arm position relative to the most proximal tracked point colored by time from substrate release of the arm. (**c**) Curvature and (**d**) strain of arm L2 over time.

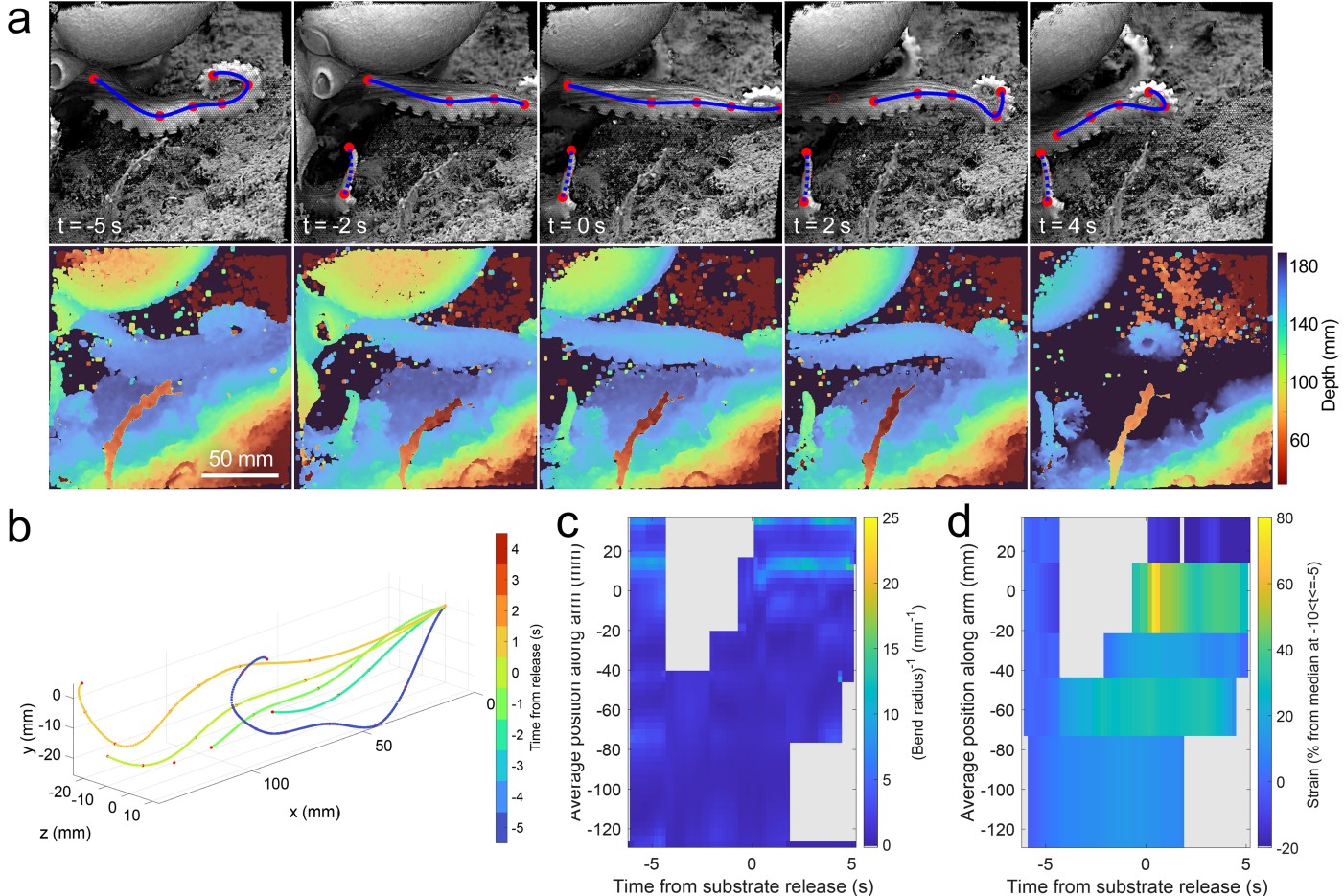

**Extended Data Fig. 4 | Representative data of the detailed kinematics of arm L4 of individual O15. (a)** Total focus images at various times relative to release of the contact point, with tracked points overlaid (top row) and depth maps (bottom row). **(b)** 3D plot of arm position relative to the most proximal tracked point colored by time from substrate release of the arm. **(c)** Curvature and **(d)** strain of arm L4 over time.

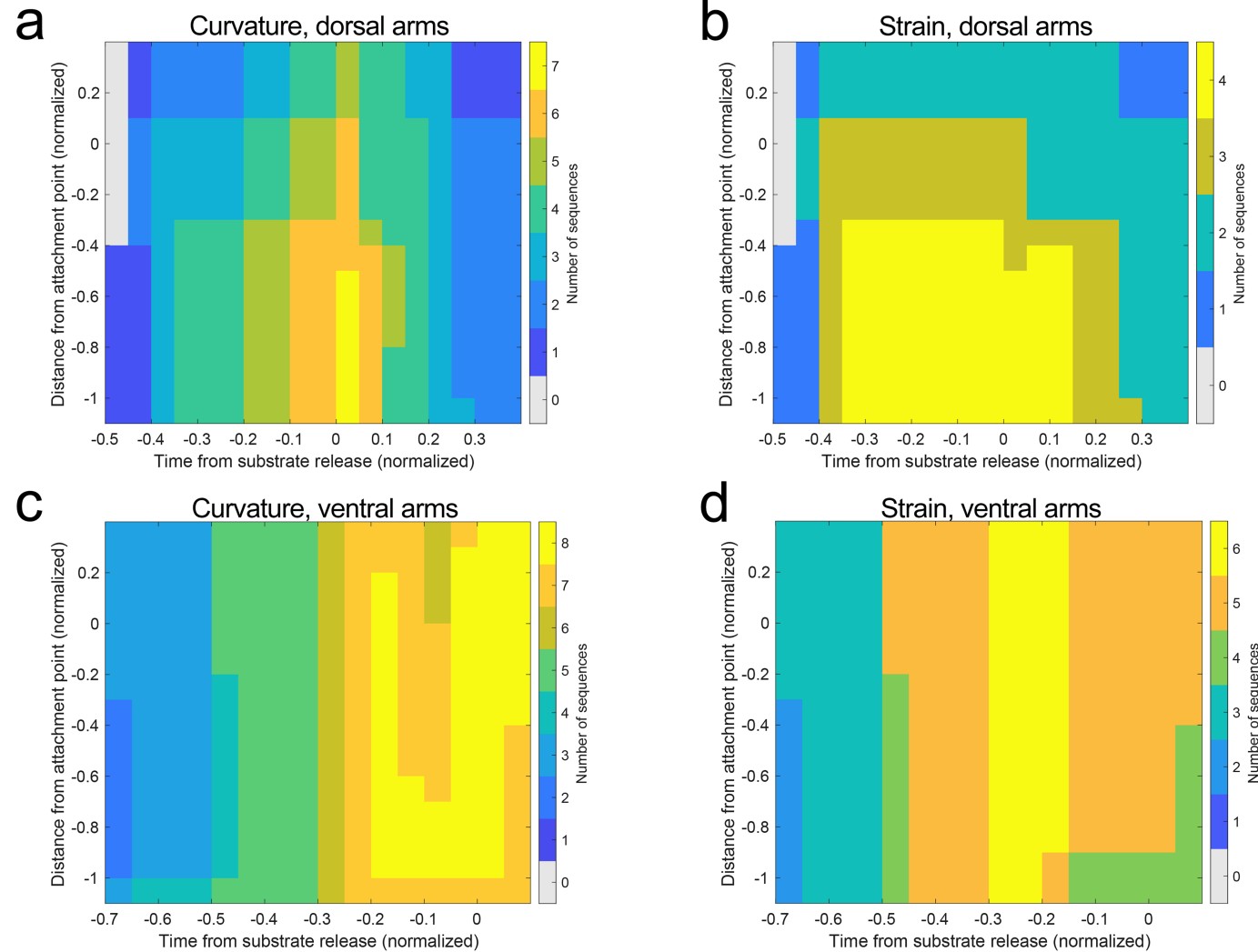

**Extended Data Fig. 5 | Number of sequences used for aggregate kinematics.** Number of octopus crawling arm cycles in each time-position bin that are used to inform the aggregate bend radius and curvature values shown in Fig. 5.

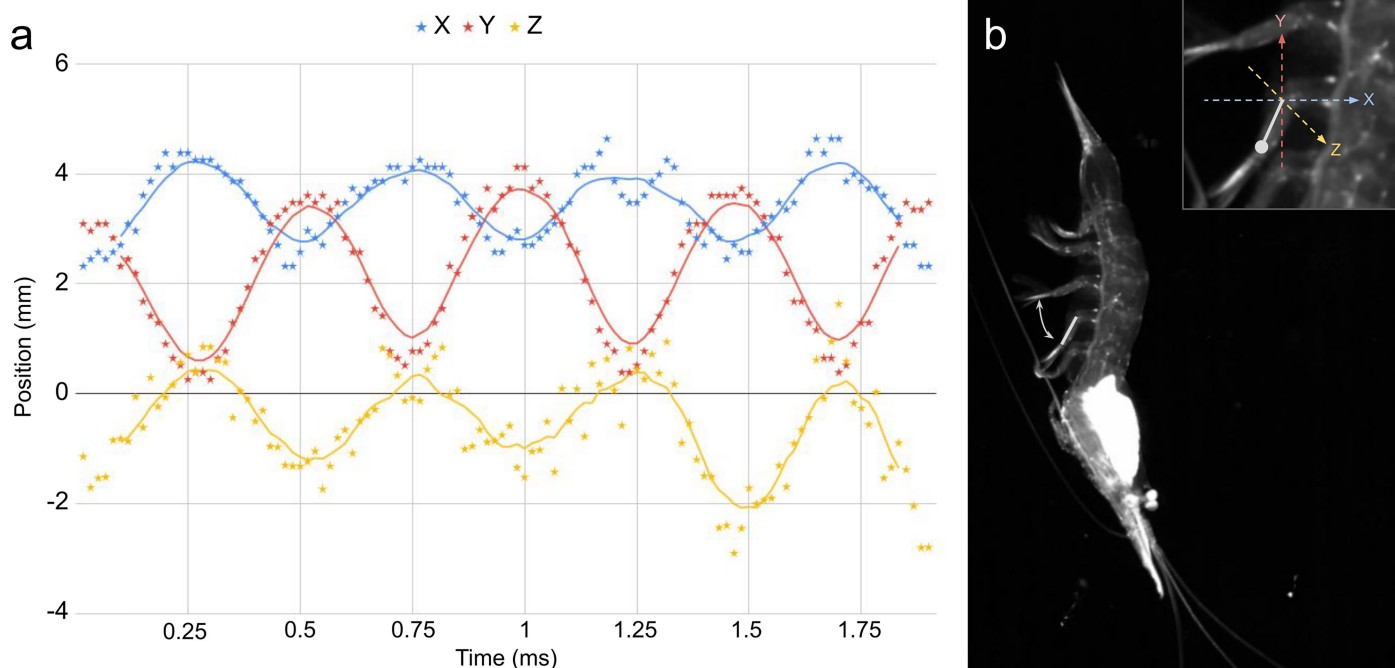

**Extended Data Fig. 6 | Kinematics data collected by EyeRIS can also be used to inform midwater animal biomechanics. (a)** Kinematics data of a swimming midwater pasiphaeid shrimp (**b**; reference axes are shown) was recorded using *EyeRIS* during a separate field expedition on Dec 6 2022. Two protopod joints (the origin and distal) were tracked in the total focus orthographic projection view and used to compute the three dimensional displacement relative to the joint origin at 60 frames per second. Sinusoidal displacements in all three axes (x-axis is blue; y-axis is red; z-axis is yellow) are clearly resolved using this approach. Solid lines are the "gam" smoothing lines.

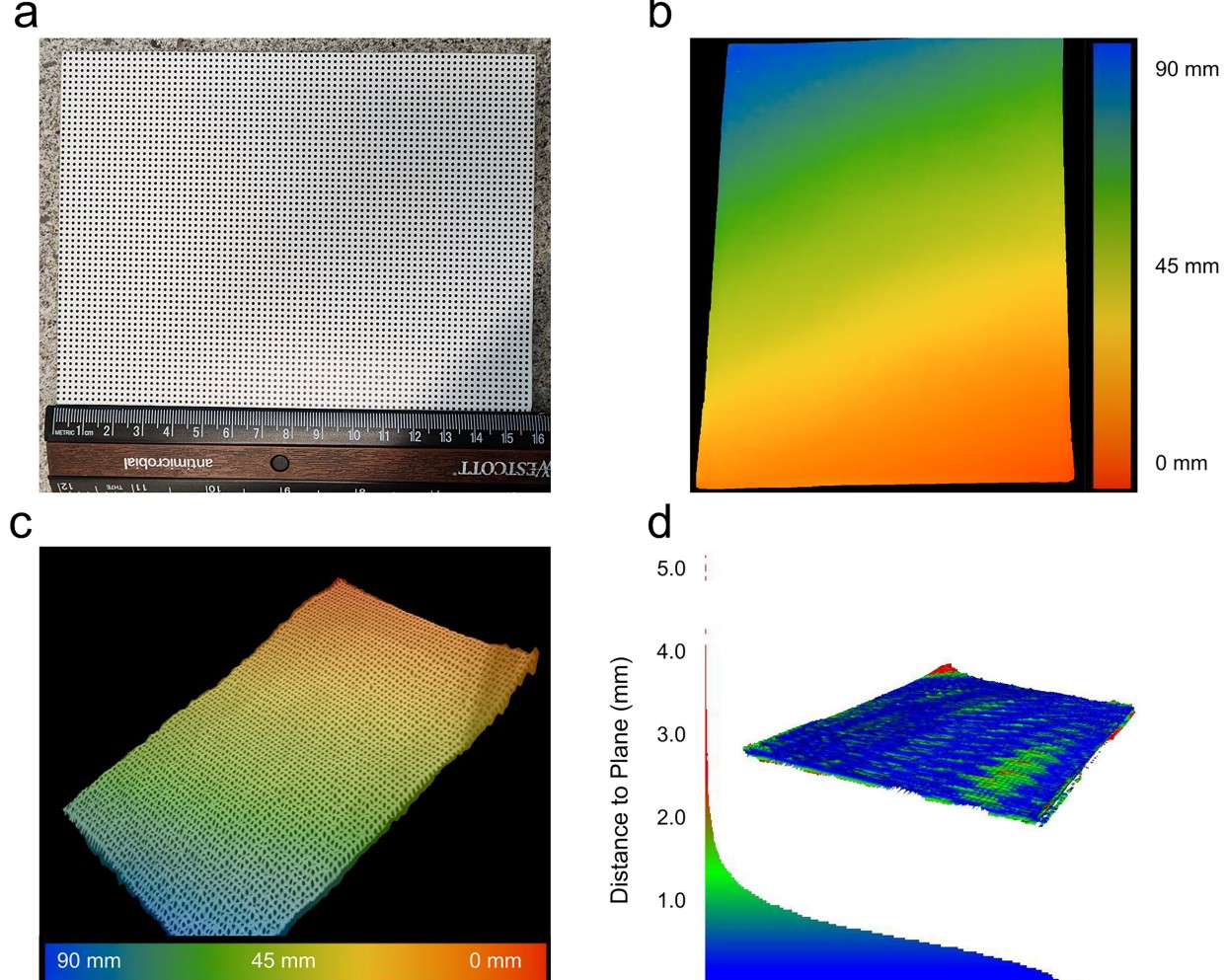

**Extended Data Fig. 7 | In-water validation process showing the three-dimensional measurement capabilities of *EyeRIS*. (a)** Calibration target is a 160 mm x 120 mm aluminum-printed board with a dot pattern of 1 mm dots on a 2 mm grid. **(b)** Depth map of the calibration target, where colors indicate the distance from the nearest-resolved plane. **(c)** Three-dimensional reconstruction and resulting mesh model of the calibration target with the depth map overlaid as a texture element. **(d)** Comparison of the derived mesh model with the known mesh using one-sided Hausdorff distance, where warm colors indicate larger distortions. In the mesh model comparisons, a simple 4-point mesh alignment was performed without any scaling of the mesh; the alignment was not optimized to minimize any error. Error measurements at various positions within the imaged volume are shown in Extended Data Table 1.

**Extended Data Table 1 | Summary of point-to-point error measurements during validation of the *EyeRIS* system (see Extended Data Fig. 7 for the calibration target)**

| x1 (mm) | y1 (mm) | z1 (mm) | x2 (mm) | y2 (mm) | z2 (mm) | Measured Dist. (mm) | Known Dist. (mm) | Error (mm) | Error (%) |
|---------|---------|---------|---------|---------|---------|---------------------|------------------|------------|-----------|
| -50.97 | -69.76 | 38.76 | 38 | -58.22 | 112.44 | 116.1 | 116 | 0.1 | 0.086 |
| -6.29 | -46.48 | 72.96 | -6.58 | 60.48 | 56.64 | 108.21 | 108 | 0.21 | 0.194 |
| -39.35 | 69.69 | 26.95 | -39.29 | 59.77 | 28.12 | 9.99 | 10 | 0.01 | 0.100 |
| 31.38 | -41.82 | 105.69 | 31.41 | -51.62 | 107.31 | 9.93 | 10 | 0.07 | 0.700 |
| -43.06 | -65.01 | 44.38 | -41.52 | -64.91 | 45.56 | 1.93 | 2 | 0.07 | 3.500 |
| 27.01 | 66.59 | 85.76 | 28.45 | 66.81 | 87.03 | 1.94 | 2 | 0.06 | 3.000 |

The error was derived from measuring different lengths at different locations within the imaged volume. The coordinates of the points are shown in the first six columns followed by the measured distance, known distance, absolute error, and relative error.

# Reporting Summary

## Statistics

For all statistical analyses, confirm that the following items are present in the figure legend, table legend, main text, or Methods section.

| n/a | Confirmed | |
|---|---|---|
| ☐ | ☒ | The exact sample size (*n*) for each experimental group/condition, given as a discrete number and unit of measurement |
| ☐ | ☒ | A statement on whether measurements were taken from distinct samples or whether the same sample was measured repeatedly |
| ☒ | ☐ | The statistical test(s) used AND whether they are one- or two-sided *Only common tests should be described solely by name; describe more complex techniques in the Methods section.* |
| ☒ | ☐ | A description of all covariates tested |
| ☒ | ☐ | A description of any assumptions or corrections, such as tests of normality and adjustment for multiple comparisons |
| ☐ | ☒ | A full description of the statistical parameters including central tendency (e.g. means) or other basic estimates (e.g. regression coefficient) AND variation (e.g. standard deviation) or associated estimates of uncertainty (e.g. confidence intervals) |
| ☒ | ☐ | For null hypothesis testing, the test statistic (e.g. *F*, *t*, *r*) with confidence intervals, effect sizes, degrees of freedom and *P* value noted *Give P values as exact values whenever suitable.* |
| ☒ | ☐ | For Bayesian analysis, information on the choice of priors and Markov chain Monte Carlo settings |
| ☒ | ☐ | For hierarchical and complex designs, identification of the appropriate level for tests and full reporting of outcomes |
| ☒ | ☐ | Estimates of effect sizes (e.g. Cohen's *d*, Pearson's *r*), indicating how they were calculated |

*Our web collection on statistics for biologists contains articles on many of the points above.*

## Software and code

Policy information about availability of computer code

| Data collection | General instrument control and operation software is provided through open-source repositories reported in the software policy. Proprietary software RxLive 6.0 by Raytrix was used to adjust and save camera sensor information, and derive spatial scales. |
|---|---|
| Data analysis | RxLive 6.0 was also used to export 3D-reference imagery, which was then analyzed by custom analysis code run in MATLAB R2022b. This code is provided publicly in a GitHub repository. |

For manuscripts utilizing custom algorithms or software that are central to the research but not yet described in published literature, software must be made available to editors and reviewers. We strongly encourage code deposition in a community repository (e.g. GitHub). See the Nature Portfolio guidelines for submitting code & software for further information.

## Data

Policy information about availability of data

All manuscripts must include a data availability statement. This statement should provide the following information, where applicable:

- Accession codes, unique identifiers, or web links for publicly available datasets
- A description of any restrictions on data availability
- For clinical datasets or third party data, please ensure that the statement adheres to our policy

All image data used for the analysis described in this manuscript, as well as derived, processed data, is provided on Zenodo at 10.5281/zenodo.10795493

# Research involving human participants, their data, or biological material

Policy information about studies with <u>human participants or human data</u>. See also policy information about <u>sex, gender (identity/presentation), and sexual orientation</u> and <u>race, ethnicity and racism</u>.

| | |
|---|---|
| Reporting on sex and gender | Not applicable. |
| Reporting on race, ethnicity, or other socially relevant groupings | Not applicable. |
| Population characteristics | Not applicable. |
| Recruitment | Not applicable. |
| Ethics oversight | Not applicable. |

Note that full information on the approval of the study protocol must also be provided in the manuscript.

# Field-specific reporting

Please select the one below that is the best fit for your research. If you are not sure, read the appropriate sections before making your selection.

☐ Life sciences          ☐ Behavioural & social sciences          ☒ Ecological, evolutionary & environmental sciences

For a reference copy of the document with all sections, see <u>nature.com/documents/nr-reporting-summary-flat.pdf</u>

# Ecological, evolutionary & environmental sciences study design

All studies must disclose on these points even when the disclosure is negative.

| | |
|---|---|
| Study description | Specialized cameras were used to quantify octopus kinematics and locomotion characteristics in-situ at over 3000 meter depth using a remotely operated vehicle (ROV). |
| Research sample | The research sample consisted of all kinematics data collected of crawling Muusoctopus robustus at the deepwater site at Davidson Seamount in a single dive of a remotely operated vehicle. During the dive, data collection occurred on any animals that were actively crawling upon first observation, and showed minimal if any disturbance behavior upon approach (such as hiding in the substrate). Considerations were given to approach the animal with minimal disturbance to the sea floor and other animal life. |
| Sampling strategy | Animal sex was determined but not used for decision on inclusion in the research sample. As described above, we collected data on all actively crawling individuals within the time frame of the ROV dive. |
| Data collection | Data was collected by cameras that recorded the entire ROV dive, as well as our imaging instrument that was manually triggered to record video when the animal of interest was framed well and displayed natural behavior. |
| Timing and spatial scale | Limited time at depth constrained all data collection within a timeframe of approximately 6 hours, with opportunistic sampling based on observed animal behavior. |
| Data exclusions | Only data where no 3D data could be extracted was excluded. |
| Reproducibility | Access limitations precluded us from repeating the sampling on a different day, but behaviors of octopus in archival footage (without 3D measurements) appear to be comparable to those observed in our sample. |
| Randomization | Due to the large population of octopus at the site, and the natural variation in behaviors, we believe our opportunistic sampling should represent a random sample. |
| Blinding | No blinding was performed, but sample information such as size and sex was determined from video footage and 3D data after the kinematics analysis had taken place. |

Did the study involve field work?          ☒ Yes          ☐ No

## Field work, collection and transport

| | |
|---|---|
| Field conditions | Deep sea Octopus Gardens located on Davidson Seamount in the Monterey Bay National Marine Sanctuary |

| Location | 35.518911∘N, 71 122.64085∘W; ∼ 3230 m deep |
|---|---|
| Access & import/export | The work was supported by a California Department of Fish and Wildlife license (to KK) and a general use permit for the Monterey Bay National Marine Sanctuary (to MBARI). |
| Disturbance | When landing the observational vehicle on the seafloor, there is potential for damaging biota. To minimize these disruptions, we would only study animals in areas that were free from other potentially impacted biota. |

# Reporting for specific materials, systems and methods

We require information from authors about some types of materials, experimental systems and methods used in many studies. Here, indicate whether each material, system or method listed is relevant to your study. If you are not sure if a list item applies to your research, read the appropriate section before selecting a response.

## Materials & experimental systems

| n/a | Involved in the study |
|---|---|
| ☒ | Antibodies |
| ☒ | Eukaryotic cell lines |
| ☒ | Palaeontology and archaeology |
| ☐ | ☒ Animals and other organisms |
| ☒ | Clinical data |
| ☒ | Dual use research of concern |
| ☒ | Plants |

## Methods

| n/a | Involved in the study |
|---|---|
| ☒ | ChIP-seq |
| ☒ | Flow cytometry |
| ☒ | MRI-based neuroimaging |

## Animals and other research organisms

Policy information about studies involving animals; ARRIVE guidelines recommended for reporting animal research, and Sex and Gender in Research

| Laboratory animals | No laboratory animals were used in this study. |
|---|---|
| Wild animals | We studied Muusoctopus robustus (octopuses) in the field and did not collect them for further observations. We could not determine age of individuals. |
| Reporting on sex | Individuals of both sexes were observed crawling or swimming at the site. The sex of individual octopuses were determined by watching multiple perspectives and identifying the easily-recognizable mating arm of males. |
| Field-collected samples | No wild animals were used in this study. |
| Ethics oversight | No ethical approval was required since animals were studied non-invasively in the wild. |

Note that full information on the approval of the study protocol must also be provided in the manuscript.

## Plants

| Seed stocks | Not applicable. |
|---|---|
| Novel plant genotypes | Not applicable. |
| Authentication | Not applicable. |

