## [Peer Review file · Nature]

In situ lightfield imaging of octopus locomotion reveals simplified control

Corresponding Author: Dr Kakani Katija

Version 0:

Reviewer comments:

Referee #1

(Remarks to the Author)

The paper reports on the use of EyeRIS – a custom-built in situ lightfield imaging system – for field observations and analysis of octopus movement. Field data was collected from a deep-sea environment known to have a relatively high abundance of octopi. Impressive data with interesting analysis, particularly discussions on the strain and gait analysis seems valuable. The authors motivate the work by pointing out the current paucity in “real-world” observations of octopus locomotion – the current dataset reported here, while limited, present some pretty interesting (and purportedly the first such documentation), with relevance to biomimetic engineering design. The manuscript is well-written, structured appropriately and is easy to follow and read.

Below are some broad comments and suggestions for improving the manuscript:

1. The title seems a little off. The first word says EyeRIS (the system), but the second part (post the colon) refers to measurements and science (and not the system). Maybe reconsider wording it? E.g., “EyeRIS: In situ lightfield imaging system reveals...” or something which refers to the system, or drop the “EyeRIS” altogether from the title.
2. Abstract last line: “...will enable us to mine the ocean for biological inspiration in the deep ocean.” Repetitive use of ocean, maybe rephrase?
3. Data availability: While the authors state: “All data, code, and materials used in this work can be accessed in public repositories at 62,63, and57, 58, respectively.”, the links are either not available or access to data denied (e.g., see Zenodo link in ref 62). Refs 57-58 and 63 (as listed in the manuscript) refer to 'MBARI report' - which is not readily accessible, a url would be helpful. While the authors did provide this upon request for the review process, this needs to be addressed in the final manuscript.
4. As the new instrument is being introduced for the first time (unless I have missed a prior publication), it would be nice to discuss the instrumentation a little more in terms of the following: What are the primary challenges and drawbacks that are anticipated in deployments or where the instrument will not work? For example, would this be useful in highly dynamic, turbid, coastal environments for similar observations? Are observations limited to the benthic region? Etc.
5. Movie ED2 – the caption refers to (a), (b), (c) – but the movie does not have the labels (and also there are 4 sub-videos, so a-d). Please add.
6. Line 104: “The order of arms used in crawling (Fig. 3F) shows a higher occurrence of cyclical contact and weight-bearing from arms 4 to 3 to 2 and back again to 4, with little to no occurrence of arms 1 following a consistent arm pair.” The figure appears to cycle from L3 to R3 to R2 to L3?
7. Figure 5 show that the peak strain occurs prior to release for ventral arms and after for dorsal. Does this support that the ventral arms are primed for weight bearing and dorsal arms for sensing?
8. Line 152 : It was not entirely clear as to what “punting” means in the context of octopus locomotion. Would be good to elaborate briefly.
9. Line 322: What is the field of view at chosen lens settings? Since for light field imaging multiple pixels are recording the same patch at various angles, what is the effective resolution?
10. Line 333: Was 100 Mb connection sufficient to get real time feed at 60 fps or was there any significant lag experienced?

11. Line 358: What would be the typical uncertainty in obtaining depth maps and total focus images? The focused images seem to capture the animal skin texture, with noticeable difference in appearance under strained and relaxed conditions. Could the authors provide their thoughts on the ability to use this textural information quantitatively?
12. Line 372: How sensitive is the calibration to changes in refractive index of the seawater?
13. Line 392: Was the depth map used? Could you comment on ease of automating this process?
14. Line 417: How was outliers defined? Upon reviewing the code, it seems to be based on standard deviation but this should be explicitly stated in the text.
15. What was the typical movement of 3D tracked points between frames?
16. Could you discuss how sensitive the presented imaging technology would be to distortions due to refraction at the window, turbulence, etc. Would the median filtering, peak fitting, etc. remove these effects in the subsequent analysis without any bias?
17. Figure ED1/ED7 - the subplot labels (a,b) are small case while everywhere else uppercase is used (A,B etc). Figure ED6 needs subplot labels (A,B)

(Remarks on code availability)

The shared code could be better commented, with instructions to download the dependencies such as `nanmedfilt2`, `optimizeFig`, curve fitting and parallel computing toolboxes etc.

The present code also expects a 3D array for the depth maps, whereas the shared data is 2D with a reference 3D image. This makes sense for memory saving, however the code needs to account for this.

Referee #2

(Remarks to the Author)

I read the manuscript entitled "EyeRIS: Three-dimensional in situ measurements of deep-sea octopus locomotion reveal simplified control," by Katija et al.

The manuscript presents a new technology for deep sea observation, consisting of a lightfield imaging system EyeRIS and an ultra-high-definition science camera. The manuscript shows the capabilities of this technology by reporting new data on octopus locomotory behavior in situ, in a recently discovered 2000-meter-deep Octopus Garden off the coast of California. Crawling, swimming, brooding, and stationary behaviors are observed in a total of 15 octopus for various time durations and various fields of view.

This new technology is a feat of engineering! It integrates advanced cameras and cutting-edge imaging technology, leveraging state-of-the-art image analysis and 3D reconstruction techniques, for capturing and video recording 3D images of deep ocean animals, here the octopus *Mussoctopus robustus*. Supplementary Movie 1 is an engineering achievement in its own right.

I am very enthusiastic about the eyeRIS system and, more generally, about the broader research project led by Dr. Katija, to enable scientific measurements in the wild, and not just any wild, but difficult to reach places in the deep ocean. However, from a scientific perspective, I have concerns about the logic presented throughout the manuscript, suggesting that lab experiments are overly restrictive, intrusive, and not fully representative of animal behavior in natural environments. This may be true for certain behavior but perhaps not as much of locomotory motives. For example, a fish in an aquarium swims like its cousin in the ocean, a domestic cat walks like its wild relative, and an incarcerated person walks, well, like a person. The paucity of scientific studies of octopus locomotory patterns may be attributed to the scientists themselves, and not to what the octopus is or isn't capable of doing in captivity. The latter is hard to pin down. Case in point, if the same dataset were obtained from 3D image reconstruction and analysis of a lab octopus, we wouldn't be discussing it as candidate for publication in Nature.

Along the same lines, the deep-ocean robotic platform carrying the eyeRIS system and the accompanying illumination needed for imaging in the dark deep ocean are by themselves a large perturbation to the environment that these organisms are accustomed to. This large perturbation may be as severe as any lab manipulation, especially to an organism with eyes and a large brain like the octopus.

The main findings about the locomotory patterns of *M. Robustus* in the wild, including that the arm bends in distinct regions, forming joint-like elements, have been reported in the literature before, as summarized beautifully in lines 140-156.

The statement that the crawling patterns show elements of simplified but flexible control is not sufficiently substantiated. This is the main thesis of the work -- it appears in the title -- yet the reported data is not sufficient to reach this strong conclusion.

The manuscript is well written and relatively easy to follow. As a scientist, I did not enjoy the dual thread of focusing on the

technology itself and the observations in the octopus. The manuscript read more like a technical description than a presentation of radically new science.

In my opinion, the contribution of this work is primarily technological rather than scientific. Yet, it has a strong appeal: the deep ocean and the octopus, an organism that has captured much of the imagination of the public. I leave the decision of whether this is sufficient for publication in Nature to the Editor.

(Remarks on code availability)

Version 1:

Reviewer comments:

Referee #1

(Remarks to the Author)

The authors have satisfactorily addressed all of our comments in the rebuttal. The only comment is that they could also have included 1-2 sentences regarding responses to our points #7,9, 13 and 15 in the text of the manuscript at relevant locations as they might be helpful in addressing questions readers might have (but maybe they are restricted by word limits?). We leave it to their discretion to include in the final version.

(Remarks on code availability)

Referee #2

(Remarks to the Author)

I read the revised manuscript.

A. As in the first version, the strengths of the revised manuscript are: (1) the robotic system eyeRIS for deep ocean scientific imaging and monitoring of animal behavior in their habitat; and (2) using this system to image the octopus *Musoctopus robustus* in a natural setting in the Garden on 56 Davidson Seamount (California, USA, 3200 m deep).

B. The work is significant! The robotic and imaging system is a feat of engineering.

C. The data and methodology concerning the robotics, imaging and image analysis are thorough and of high quality. The leap from this data to drawing conclusions on the neuromechanical "control" of these animals is not scientifically rigorous.

D. No comment

E. Same comment as in C: The statements on the simplified neuromechanical "control" of the octopus are not scientifically rigorous; they are hypotheses drawn from time-history of morphological arm deformations -- from the presented data, we cannot tell if the neuromechanical control of these arms movements is simple or not.

F. To conclude that the octopus uses simplified neuromechanical "control" for its locomotion, it is not enough to look at morphological data of the arms. One would need complementary methods from neuromechanics and/or mathematical modeling (feedback control).

G. Seems appropriate

H. The writing is clear and the study is fascinating

(Remarks on code availability)

Referees' comments:

Referee #1 (Remarks to the Author):

The paper reports on the use of EyeRIS – a custom-built in situ lightfield imaging system – for field observations and analysis of octopus movement. Field data was collected from a deep-sea environment known to have a relatively high abundance of octopi. Impressive data with interesting analysis, particularly discussions on the strain and gait analysis seems valuable. The authors motivate the work by pointing out the current paucity in “real-world” observations of octopus locomotion – the current dataset reported here, while limited, present some pretty interesting (and purportedly the first such documentation), with relevance to biomimetic engineering design. The manuscript is well-written, structured appropriately and is easy to follow and read.

Below are some broad comments and suggestions for improving the manuscript:

1. The title seems a little off. The first word says EyeRIS (the system), but the second part (post the colon) refers to measurements and science (and not the system). Maybe reconsider wording it? E.g., “EyeRIS: In situ lightfield imaging system reveals...” or something which refers to the system, or drop the "EyeRIS" altogether from the title.

Thank you for this suggestion. We really struggled with the title of this manuscript. We have changed it to: “*EyeRIS*: In situ lightfield imaging of deep-sea octopus locomotion reveals simplified control.”

2. Abstract last line: “...will enable us to mine the ocean for biological inspiration in the deep ocean.” Repetitive use of ocean, maybe rephrase?

Thank you for bringing this to our attention. We have modified the text, which now reads: “... will enable mining of the deep ocean for biological inspiration.”

3. Data availability: While the authors state: "All data, code, and materials used in this work can be accessed in public repositories at 62,63, and57, 58, respectively.", the links are either not available or access to data denied (e.g., see Zenodo link in ref 62). Refs 57-58 and 63 (as listed in the manuscript) refer to 'MBARI report' - which is not readily accessible, a url would be helpful. While the authors did provide this upon request for the review process, this needs to be addressed in the final manuscript.

Apologies for this inconvenience. The citation format we used for the submission obscured the url for the materials you referenced above. We will work with the editorial office to ensure that these urls are exposed in the final manuscript. We also checked to make sure access to the data were not denied.

4. As the new instrument is being introduced for the first time (unless I have missed a prior publication), it would be nice to discuss the instrumentation a little more in terms of the following: What are the primary challenges and drawbacks that are anticipated in deployments or where the instrument will not work? For example, would this be useful in highly dynamic, turbid, coastal environments for similar observations? Are observations limited to the benthic region? Etc.

Thank you for this suggestion. EyeRIS has been described previously in another manuscript (Burns et al, *Science Advances* 2024), and there the focus was on morphological sizing. We added a sentence (lines 187-189) to address your other questions: “Although this imaging approach would be significantly challenged in highly turbid environments, with appropriate illumination and housing hardware, it could be used to inform a number of areas of inquiry from near-surface to deep-sea environments.”

5. Movie ED2 – the caption refers to (a), (b), (c) – but the movie does not have the labels (and also there are 4 sub-videos, so a-d). Please add.

Thank you for bringing this to our attention. We have modified the movie to include the subpanel labels.

6. Line 104: “The order of arms used in crawling (Fig. 3F) shows a higher occurrence of cyclical contact and weight-bearing from arms 4 to 3 to 2 and back again to 4, with little to no occurrence of arms 1 following a consistent arm pair.” The figure appears to cycle from L3 to R3 to R2 to L3?

Thank you for bringing this to our attention, and we do agree. We’ve modified the text on line 104 to say: “... weight-bearing from ventral to dorsal arms, with little to no occurrence of arms 1 following a consistent arm pair.”

7. Figure 5 show that the peak strain occurs prior to release for ventral arms and after for dorsal. Does this support that the ventral arms are primed for weight bearing and dorsal arms for sensing?

Yes, indeed. This supports the idea that ventral arms are primed for weight bearing and dorsal arms for sensing.

8. Line 152 : It was not entirely clear as to what “punting” means in the context of octopus locomotion. Would be good to elaborate briefly.

We don't think “punting” adds clarity to describe the process so it has been removed from the manuscript.

9. Line 322: What is the field of view at chosen lens settings? Since for light field imaging multiple pixels are recording the same patch at various angles, what is the effective resolution?

Effective resolution varies over the depth of the imaged volume, but is maximal at one quarter of the sensor resolution (in this case 6.25 MP). For a field of view of 120 x 120 mm, spatial resolution would be 48 um.

10. Line 333: Was 100 Mb connection sufficient to get real time feed at 60 fps or was there any significant lag experienced?

Camera control and data utilized a fiber connection to get data rates up to 12 Gb/s that enabled viewing and recording data at 60 fps, which was sufficient for real-time observations with minimal lag. Instrument control utilized a 100 Mb/s ethernet connection. We added text on lines 350-352 for clarity.

11. Line 358: What would be the typical uncertainty in obtaining depth maps and total focus images? The focused images seem to capture the animal skin texture, with noticeable difference in appearance under strained and relaxed conditions. Could the authors provide their thoughts on the ability to use this textural information quantitatively?

Typical uncertainty in depth maps are less than 1 mm (Figure ED7,d). Uncertainty in total focus texture is significantly lower, and limited by the effective spatial resolution of the lightfield camera, or roughly one quarter of the total number of pixels of the sensor. We would expect the system to provide excellent texture reproduction of features on the order of millimeters. We added lines 398-399 to make that explicit: “Typical depth maps using EyeRIS in this study have an uncertainty that is less than 1 mm (Fig. ED7,d).”

12. Line 372: How sensitive is the calibration to changes in refractive index of the seawater?

The calibration procedure for the Raytrix camera is minimally sensitive to changes in seawater refractive index. This is because the microlens array calibration and depth estimation is performed in virtual depth (image-space) and does not rely on object-space measurements (this step can be performed in air or other media, for example). A metric calibration is still required to map virtual depth to physical depth, but this calibration is only correcting for lens decenter and tilt and is not strongly sensitive to focal length. We added an explanation in lines 391-392: "It should be noted that typical variations in the refractive index of seawater due to pressure, temperature, and salinity do not significantly change the calibration."

13. Line 392: Was the depth map used? Could you comment on ease of automating this process?

For the whole-body gait analysis, the depth map was used only in cases where the total focus and other perspective camera views were insufficient to confirm whether an arm was touching the substrate. Due to continuously changing orientation of the animal with respect to our instrumentation, human annotation of touching / not touching was required, and automation would be challenging. However, to speed up the analysis performed for this manuscript in general, we can envision the use of the depth maps to better constrain the point tracking in the detailed arm kinematics, where the current method required significant manual intervention.

14. Line 417: How were outliers defined? Upon reviewing the code, it seems to be based on standard deviation but this should be explicitly stated in the text.

We agree with the reviewer that this should be stated, and have added the following to lines 440-442: "It's worth noting that for outlier correction, a baseline was generated by applying a moving median width of 50 time steps (0.83 s) to the z-coordinates. Any point more than the greater of 5 mm or one standard deviation from the reference was removed." The reviewer correctly located and interpreted the relevant code, which we respect and appreciate.

15. What was the typical movement of 3D tracked points between frames?

Typical movement of 3D tracked points between adjacent frames was less than 5 pixels. DLTdv8 provides sub-pixel tracking, and the described temporal smoothing was intended to improve our confidence in the derived values.

16. Could you discuss how sensitive the presented imaging technology would be to distortions due to refraction at the window, turbulence, etc. Would the median filtering, peak fitting, etc. remove these effects in the subsequent analysis without any bias?

As the lightfield depth estimation algorithm operates within microlens sub apertures with narrow fields of view, it is minimally impacted by flat port distortion at the pressure window. Steady-state radial distortion due to deformation of the pressure window or bubble-entrainment would be problematic for this method (as for any other 3D imaging method). In contrast, time-varying distortions would in most cases be filtered out and not add bias to the subsequent analysis.

17. Figure ED1/ED7 - the subplot labels (a,b) are small case while everywhere else uppercase is used (A,B etc). Figure ED6 needs subplot labels (A,B)

From the Nature guidelines for authors, the appropriate subpanel formatting is lowercase letters, so we have fixed this throughout the manuscript.

Referee #1 (Remarks on code availability):

The shared code could be better commented, with instructions to download the dependencies such as nanmedfilt2, optimizeFig, curve fitting and parallel computing toolboxes etc. The present code also expects a 3D array for the depth maps, whereas the shared data is 2D with a reference 3D image. This makes sense for memory saving, however the code needs to account for this.

Thank you for pointing this out. We included optimizeFig and added to the installation instructions and system requirements. Indeed the depth map files on the public repository were streamlined for accessibility (other image channels were repeated in each frame). We have modified the code to be compatible with that format, and allow for easier execution on an example subset of the data. To that end, we added a script with instructions to reproduce the analysis for this subset, and recreated several of the figures in our manuscript with minimal user input. We were careful not to change code functionality from the state used for our analyses, but did introduce some refactoring for ease of use.

Referee #2 (Remarks to the Author):

I read the manuscript entitled “EyeRIS: Three-dimensional in situ measurements of deep-sea octopus locomotion reveal simplified control,” by Katija et al.

The manuscript presents a new technology for deep sea observation, consisting of a lightfield imaging system EyeRIS and an ultra-high-definition science camera. The manuscript shows the capabilities of this technology by reporting new data on octopus locomotory behavior in situ, in a recently discovered 2000-meter-deep Octopus Garden off the coast of California. Crawling, swimming, brooding, and stationary behaviors are observed in a total of 15 octopus for various time durations and various fields of view.

This new technology is a feat of engineering! It integrates advanced cameras and cutting-edge imaging technology, leveraging state-of-the-art image analysis and 3D reconstruction techniques, for capturing and video recording 3D images of deep ocean animals, here the octopus *Musoctopus robustus*. Supplementary Movie 1 is an engineering achievement in its own right.

Thank you for saying this. We really appreciate it.

I am very enthusiastic about the eyeRIS system and, more generally, about the broader research project led by Dr. Katija, to enable scientific measurements in the wild, and not just any wild, but difficult to reach places in the deep ocean. However, from a scientific perspective, I have concerns about the logic presented throughout the manuscript, suggesting that lab experiments are overly restrictive, intrusive, and not fully representative of animal behavior in natural environments. This may be true for certain behavior but perhaps not as much of locomotory motives. For example, a fish in an aquarium swims like its cousin in the ocean, a domestic cat walks like its wild relative, and an incarcerated person walks, well, like a person. The paucity of scientific studies of octopus locomotory patterns may be attributed to the scientists themselves, and not to what the octopus is or isn't capable of doing in captivity. The latter is hard to pin down. Case in point, if the same dataset were obtained from 3D image reconstruction and analysis of a lab octopus, we wouldn't be discussing it as a candidate for publication in *Nature*.

Thank you for this comment, and you make a good point about locomotory motives potentially being captured sufficiently within a laboratory environment.

We've made several changes to the text to soften our language in response to your comment here, which include:

- 1. Changed lines 32-33: “can provide important perspectives on how these animals effectively operate their complex bodies.”

2. Changed lines 37-39: “While there are practical reasons for conducting observations in the laboratory, free-living individuals must simultaneously negotiate a wider variety of conditions, including unpredictable changes in terrain, water currents, and interactions with other animals.”
3. Changed lines 41-42: “The ability to conduct quantitative studies using imaging in unconstrained octopuses in uncontrolled situations is an important step toward understanding their entire movement repertoire.”
4. Changed lines 172-174: “Observations presented here, by contrast, were conducted in the wild, where the unconstrained scale and variability of conditions allowed free-living octopuses to express a broader range of simultaneous control solutions.”

It's worth noting that uncontrolled observations (like those we did here in situ) show us things we haven't thought to test yet in the lab (i.e., bipedal locomotion in octopuses was first observed in the wild). Finally, quantitative 3D image reconstructions of octopus locomotion, even in a lab environment, would still represent a significant step forward for the field without the additional element of in situ observations.

Along the same lines, the deep-ocean robotic platform carrying the eyeRIS system and the accompanying illumination needed for imaging in the dark deep ocean are by themselves a large perturbation to the environment that these organisms are accustomed to. This large perturbation may be as severe as any lab manipulation, especially to an organism with eyes and a large brain like the octopus.

Indeed octopuses can be sensitive to lights and being followed by an individual observer or platform. We wish we could reference a study that describes in detail the habituation of wild octopuses to observer presence, but alas we are not aware of one. We have added the following on lines 83-85: “As with prior studies presenting ROV observations of deep-sea octopuses, the presence of bright lights is acknowledged, but behaviors observed are considered to be part of the animals' typical repertoire (Voight, 2008). Results presented here do not include overtly evasive maneuvers or fast escape.”

The main findings about the locomotory patterns of *M. Robustus* in the wild, including that the arm bends in distinct regions, forming joint-like elements, have been reported in the literature before, as summarized beautifully in lines 140-156. The statement that the crawling patterns show elements of simplified but flexible control is not sufficiently substantiated. This is the main thesis of the work -- it appears in the title -- yet the reported data is not sufficient to reach this strong conclusion.

We added more text (lines 131-134 and paragraph starting at line 149) that explains our reasoning and how conclusions have been drawn about simplified control in octopus locomotion in prior work. Incorporating this additional information helps strengthen our findings, so thank you for highlighting this point.

The manuscript is well written and relatively easy to follow. As a scientist, I did not enjoy the dual thread of focusing on the technology itself and the observations in the octopus. The manuscript read more like a technical description than a presentation of radically new science. In my opinion, the contribution of this work is primarily technological rather than scientific. Yet, it has a strong appeal: the deep ocean and the octopus, an organism that has captured much of the imagination of the public. I leave the decision of whether this is sufficient for publication in *Nature* to the Editor.

Thank you for finding our paper well written and relatively easy to follow. Coming from an engineering background, I personally find it difficult to separate scientific achievements from novel technologies built to enable them, and prefer to celebrate them simultaneously (see our other published works in *Science Advances* and *Nature* in 2017 and 2020, respectively). However, I do agree that switching of the narrative between the scientific and engineering results can be jarring, and I reviewed the manuscript to see where we might refine this. From that review, I shifted and reordered text regarding the engineering achievements toward the end of the Discussion section (starting on line 178), and led with the scientific contribution.

SUMMARY PARAGRAPH: All Nature papers begin with a fully referenced paragraph, typically no longer than 200 words, aimed at readers in other disciplines. This paragraph starts with a 2- to 3-sentence, basic introduction to the field; continues with a 1-sentence statement of the main findings starting 'Here we show' or an equivalent phrase; and finally, concludes with 2 to 3 sentences putting the main findings into general context so it is clear how the results described in the paper have moved the field forward. A downloadable, annotated example is available here.

Deep ocean animals have evolved a plethora of solutions to survive in an environment starkly different from our own. Expanding our biological observational capabilities can pay dividends in how we understand ocean system function, and knowledge about these biological systems can also contribute to the exploding field of bioinspired design. Here we present a novel lightfield imaging system, called *EyeRIS* (Remote Imaging System), which enables three-dimensional imaging and visualization on deep-diving underwater robotic vehicles. We demonstrate the utility of this instrument by studying crawling locomotion by octopuses at the 3200 m deep Octopus Garden on Davidson Seamount. With *EyeRIS*, we were able to non-invasively size organismal features, measure the 3D trajectories of different arms during crawling over rough terrain, and identified mechanisms for simplified control that can be valuable for designing octopus-inspired robotic systems. Measurements like those enabled by *EyeRIS* can transform our abilities to understand the biomechanics and behavior of marine life, and expand the horizons and available organismal models for bioinspired design research.